

# Entanglement asymmetry in periodically driven quantum systems

**Tista Banerjee, Suchetan Das and Krishnendu Sengupta⋆**

School of Physical Sciences, Indian Association
for the Cultivation of Science, Kolkata 700032, India

⋆ ksengupta1@gmail.com

## Abstract

We study the dynamics of entanglement asymmetry in periodically driven quantum systems. Using a periodically driven XY chain as a model for a driven integrable quantum system, we provide semi-analytic results for the behavior of the dynamics of the entanglement asymmetry, $\Delta S$, as a function of the drive frequency. Our analysis identifies special drive frequencies at which the driven XY chain exhibits dynamic symmetry restoration and displays quantum Mpemba effect over a long timescale; we identify an emergent approximate symmetry in its Floquet Hamiltonian which plays a crucial role for realization of both these phenomena. We follow these results by numerical computation of $\Delta S$ for the non-integrable driven Rydberg atom chain and obtain similar emergent-symmetry-induced symmetry restoration and quantum Mpemba effect in the prethermal regime for such a system. Finally, we provide an exact analytic computation of the entanglement asymmetry for a periodically driven conformal field theory (CFT) on a strip. Such a driven CFT, depending on the drive amplitude and frequency, exhibits two distinct phases, heating and non-heating, that are separated by a critical line. Our results show that for $m$ cycles of a periodic drive with time period $T$, $\Delta S \sim \ln mT \, [\ln(\ln mT)]$ in the heating phase [on the critical line] for a generic CFT; in contrast, in the non-heating phase, $\Delta S$ displays small amplitude oscillations around it's initial value as a function of $mT$. We provide a phase diagram for the behavior of $\Delta S$ for such driven CFTs as a function of the drive frequency and amplitude.

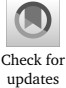

## 1 Introduction

A large variety of classical systems exhibit Mpemba effect when taken away from equilibrium [1]. This effect, which constitutes faster relaxation of a classical system when taken further away from equilibrium, is attributed to anomalous relaxation mechanism in these systems [2–5]. The Mpemba effect is now theoretically explained [2,3] and experimentally observed [4,5] in a variety of classical systems.

Initial studies aiming at understanding the quantum version of this effect constituted analysis of open quantum systems where the relaxation mechanism is implemented using a bath [6–10]. More recently, similar studies have been carried out for closed quantum systems where quench dynamics of such systems have been analyzed [11–20]. Typically, in such studies, one start from an initial state which breaks a symmetry; this is followed by the evolution of this initial state with a Hamiltonian which respects the symmetry broken by the initial state. It is found that such a time evolution leads to dynamical symmetry restoration over a typical timescale which depends on system details. In particular, these studies show a faster restoration of symmetry for initial states with larger degree of symmetry breaking leading to realization of the quantum Mpemba effect in closed systems [12].

To quantify the degree of symmetry breaking in a quantum system, one typically computes entanglement asymmetry $\Delta S$ [11–33]. Such a quantity allows one to study symmetry breaking using entanglement. Its computation begins from choosing a state of the system $|\psi\rangle$ and a subsequent construction of the density matrix $\rho = |\psi\rangle\langle\psi|$. The corresponding reduced density matrix $\rho_A$, corresponding to the subsystem $A$, is then obtained by tracing out the rest of the system (denoted as $B$ below). The entanglement entropy of this reduced density matrix can be quantified using several measures; in this work, we shall use the $n^{\text{th}}$ Renyi entropy

$$S_n = \frac{1}{1-n}\ln\operatorname{Tr}\left[\rho_A^n\right], \qquad \rho_A = \operatorname{Tr}_B\left[\rho\right]. \tag{1}$$

The next step to compute the entanglement asymmetry is to consider a symmetry $\mathcal{Q}$ with corresponding charge operator $Q$ acting on states of the system's Hilbert space and having eigenvalues $q$. This symmetry may be discrete or continuous. One can then project $\rho$ to different symmetry sectors with definite $q$; this leads to the projected density matrix

$$\rho_Q = \sum_q \Pi_q \rho \Pi_q, \qquad \Pi_q = \int_{-\pi}^{\pi} \frac{d\alpha}{2\pi} e^{i\alpha(Q-q)}, \tag{2}$$

where $\alpha$ is a real parameter. The corresponding symmetry-projected reduced density matrix and the associated $n^{\text{th}}$ Renyi entropy are denoted by $\rho_{AQ}$ and $S_{Qn} = \frac{1}{1-n}\ln\operatorname{Tr}\left[\rho_{QA}^n\right]$ respectively. If the state $|\psi\rangle$ breaks the symmetry, $\rho_A \neq \rho_{AQ}$. Using this notion, the entanglement asymmetry can be defined as

$$\Delta S_n = S_{Qn} - S_n = \frac{1}{1-n}\ln\frac{\operatorname{Tr}\left[\rho_{QA}^n\right]}{\operatorname{Tr}\left[\rho_A^n\right]}. \tag{3}$$

It is easy to see that $\Delta S_n \geq 0$ and the equality holds when $\rho_Q = \rho$ [11, 12]. This observation allows one to use $\Delta S_n$ as a probe of degree of asymmetry. The evolution of $\Delta S$ under the action of a Hamiltonian $H$ has also been studied in different contexts [11–20]. It was shown when $[H, Q] = 0$, such a evolution leads to dynamical symmetry restoration wherein $\Delta S_n \to 0$ at long times. In all the earlier works [11–20], the dynamics of symmetry restoration have been studied for quench dynamics. To the best of our knowledge, their counterpart in periodically driven integrable or non-integrable systems has not been studied so far.

The stroboscopic time evolution in a periodically driven quantum system with time period $T$ constitutes an active research area [34–38]. These driven systems are completely described by Floquet Hamiltonians $H_F$ which are related to their evolution operator $U(T, 0)$ by

$$U(mT, 0) = U^m(T, 0) = \exp\left[-iH_F(T)mT/\hbar\right], \qquad (4)$$

where $m$ is an integer. They exhibit a host of phenomena that have no analogue in either equilibrium or in the presence of aperiodic drives; these include realization of drive-induced topological phases [39–44], dynamical localization [45–49], dynamical freezing [50–55], dynamical phase transitions [56–61], and realization of time crystalline phases [62–67]. Most of the above-mentioned phenomena can be attributed to an emergent approximate symmetry of $H_F$ which shapes their nature [38]. However, the role of such an emergent symmetry on realization of Mpemba effect in a periodically driven quantum models have not been studied so far.

In this work, we study the behavior of entanglement asymmetry $\Delta S_n$ for several periodically driven quantum systems using a square-pulse protocol. For typical drive frequencies $\omega_D = 2\pi/T$, $\Delta S_n$ always remain positive showing lack of dynamical symmetry restoration. This can be understood from the fact that the Floquet Hamiltonian, at such frequencies, does not usually respect any specific symmetry; consequently, there is no dynamical symmetry restoration. However, we find that for both driven integrable and non-integrable models (such as the XY spin chain and the Rydberg chain respectively), there exists special drive frequencies for which the Floquet Hamiltonian exhibits an approximate emergent symmetry. This symmetry is emergent since it is not a property of either the system Hamiltonian or the initial state. It is approximate since it is respected only by the leading order term in $H_F$, denoted as $H_F^{(1)}$, in the high drive-amplitude regime; higher order terms, suppressed in this regime, do not respect this symmetry. Consequently, the effect of this symmetry manifests itself in these driven systems up to prethermal timescales where $H_F^{(1)}$ controls the dynamics. It is well-known that such a prethermal time scale may be exponentially large for large drive amplitudes [38, 68–70]. The presence of such a symmetry leads to dynamical symmetry restoration in such driven systems and $\Delta S_n \to 0$ at these frequencies; moreover, we show that $\Delta S_n$ exhibits Mpemba effect in the sense that its relaxation to zero occurs faster if the initial state corresponds to larger degree of symmetry breaking. To the best of our knowledge, such a realization of Mpemba effect has not been reported earlier for periodically driven integrable and non-integrable quantum systems.

We also study the behavior of $\Delta S_n$ for periodically driven conformal field theories (CFTs) defined on a strip. The properties of both symmetry projected entanglement and entanglement asymmetry has been studied earlier for CFTs in several contexts [21–28]. These works study entanglement asymmetry for the excited state of a CFT in a cylindrical geometry [22], on a semi-infinite line [23, 24], and in context of holography [25, 26]. It was shown in Refs. [21, 22] that symmetry projection for such conformal theories may be implemented via a primary vertex operator which allows one to obtain analytic expression of symmetry resolved entanglement [21] for vacuum and entanglement asymmetry for primary states [22] of the CFT on a cylinder. An analogous study for the strip geometry has not been carried out. Moreover, although periodically driven CFTs has been studied in several contexts [71–81], the behavior $\Delta S_n$ for such driven CFTs have not been studied in the literature.

To this end, we start with an initial state which manifestly breaks the symmetry generated by $Q$. A standard conformal mapping then allows us to move from the strip to the upper-half plane (UHP). This allows us to find the appearance of the conformal boundary state which breaks the symmetry corresponding to the charged symmetry sector. Since a conformal boundary state can be written as a sum of primary and Virasoro descendent states, it is expected to break the symmetry of the full theory as well as that of individual charge sectors. Thus the initial density matrix $\rho$ corresponding to such a state satisfies $[\rho, Q] \neq 0$. Consequently, the one-point functions in UHP becomes non-zero; they can be easily computed using standard method of images. In contrast, for a CFT on a cylinder, when the initial state corresponds to the CFT vacuum, $[\rho, Q] = 0$ and the entanglement asymmetry vanishes [21].

Our choice of the strip geometry also ensures that the CFT ground state, which can be thought a boundary state, evolves under a periodic drive [71]. We provide an explicit analytical expression $\Delta S_n$ for such a geometry both in equilibrium and for CFTs driven by a square-pulse protocol. It is well-known that driven CFTs, depending on the drive frequency and amplitude, support heating and non-heating phases separated by a critical line [71–81]; our analysis indicates that the behavior of $\Delta S_n$ depends crucially on the phase realized by the drive. For the heating (hyperbolic) phase, after $m \gg 1$ drive cycles, $\Delta S_n \sim \ln mT$; in contrast for the critical (parabolic) phase it shows a $\ln(\ln mT)$ growth. In the non-heating (elliptic) phase, $\Delta S_n$ is non-monotonic and exhibits small amplitude oscillation around its initial value as a function of $m$. We provide a phase diagram showing different behavior of $\Delta S_n$ as a function of the drive frequency and amplitude.

The plan of the rest of the paper is as follows. In Sec. 2, we discuss the properties of $\Delta S_n$ for an integrable XY chain showing realization of the Mpemba effect at special drive frequencies; this is followed by Sec. 3, where an analogous study is carried out for a non-integrable chain of Rydberg atoms. Next, in Sec. 4, we study $\Delta S_n$ for CFTs on a strip both in equilibrium and in the presence of a periodic drive. Finally, we discuss our main results and conclude in Sec. 5.

## 2 Driven XY chain

In this section, we study the entanglement asymmetry $\Delta S_n$ of periodically driven XY chain. For equilibrium or for a quench protocol, $\Delta S_n$ has been studied in Refs. [11–13]. The main result of the analysis of this section is the demonstration of the dynamical restoration of an approximate emergent symmetry of a periodically driven XY chain at special drive frequencies; we also find the presence of quantum Mpemba effect in such a driven chain. We show that both these features owe its existence to an approximate emergent symmetry of such driven chains at these special frequencies that have no analogue in quench protocols studied earlier.

The XY model in the presence of a time-dependent transverse magnetic field is described by the Hamiltonian

$$H_{\text{Ising}} = -\frac{J}{2}\left( \sum_{\langle j, j' \rangle} \left( \frac{1+\kappa_0}{2}\sigma_j^x \sigma_{j'}^x + \frac{1-\kappa_0}{2}\sigma_j^y \sigma_{j'}^y \right) + g(t)\sum_j \sigma_j^z \right), \tag{5}$$

where $j$ and $j'$ are coordinates of sites of the lattice in units of lattice spacing, $\langle jj' \rangle$ indicates that the sum extends over $j$ and $j'$ which are nearest neighbors and $h(t) = g(t)J$ is the transverse field which is periodically driven according to a square pulse protocol given by

$$g(t) = g_0 - (+)g_1, \quad \text{for} \quad t \leq (>)T/2. \tag{6}$$

Here $\sigma^{\alpha=x,y,z}$ denotes Pauli matrices representing the Ising spins, $\kappa_0$ is a parameter which denotes the relative strength of the interaction between $x$ and $y$ components of the spins

(with $\kappa_0 = 1$ being the Ising limit), $T = 2\pi/\omega_D$ is the time period of the drive, and $\omega_D$ is the drive frequency. In what follows, we shall set $J = 1$.

It is well-known that the XY model given by Eq. 5 can be mapped into a free fermionic model via standard Jordan-Wigner transformation relating the Ising spins to spinless fermionic fields:

$$\sigma_j^{+(-)} = \left(\prod_{i<j} -\sigma_i^z\right) c_j^\dagger(c_j), \qquad \sigma_j^z = 2c_j^\dagger c_j - 1, \tag{7}$$

where $\sigma_j^x = \sigma_j^+ + \sigma_j^-$. Substituting Eq. 7 in Eq. 5 and carrying out a subsequent Fourier transform, one obtains a free fermionic Hamiltonian in momentum space given by

$$H_{\text{ferm}} = \sum_{k>0} \psi_k^\dagger H_k \psi_k, \qquad \psi_k = (c_k, c_{-k}^\dagger)^T, \qquad H_k = \tau^z(g(t) - \cos k) + \tau^y \kappa_0 \sin k, \tag{8}$$

where $\tau^{\alpha=x,y,z}$ denotes Pauli matrices in particle-hole space.

For the square pulse protocol given by Eq. 6, the evolution operator $U(mT, 0)$ after $m$ cycles of the drive can be analytically computed. This allows one to obtain the wavefunction and hence the fermionic correlation functions of the driven chain. Using these correlators, one can follow the analysis of Refs. [11, 12] to obtain $\rho_{QA}^n$, where $\rho_{QA}$ is the symmetry resolved density matrix (Eq. 2) and $n$ is the Renyi index, after $m$ drive cycle as

$$\text{Tr}\left[\rho_{QA}^n(mT)\right] = \int_{-\pi}^{\pi} \frac{d\alpha_1..d\alpha_n}{(2\pi)^n} Z_n(\boldsymbol{\alpha}; mT),$$

$$Z_n(\boldsymbol{\alpha}; mT) = \text{Tr}\left[\prod_{j=1}^n \rho_A(mT) e^{i(\alpha_{j+1}-\alpha_j)Q}\right]. \tag{9}$$

It turns out that for integrable chains, $Z_n(\boldsymbol{\alpha}; mT)$ can be expressed in terms of the fermionic correlation matrix [11]. Since the time evolution in these integrable systems occurs independently for each $k$, the correlation matrix in the momentum space after $m$ drive cycles takes the form

$$G_k(mT) = \begin{pmatrix} \langle 2c_k^\dagger c_k - 1\rangle & -2i\langle c_{-k} c_k\rangle \\ 2i\langle c_k^\dagger c_{-k}^\dagger\rangle & \langle 2c_{-k} c_{-k}^\dagger - 1\rangle \end{pmatrix}, \tag{10}$$

where the expectations are taken with respect to the state of the chain after $m$ drive cycles.

To compute $\Delta S_n(mT)$, we therefore first obtain a semi-analytic expression for the correlation matrix of the driven chain. To this end, we note that at the end of a single drive cycle, one can write the evolution operator for any momentum mode $k$ as

$$U_k(T, 0) = e^{-iH_{kF}T/\hbar} = e^{-iH_{k+}T/(2\hbar)} e^{-iH_{k-}T/(2\hbar)}, \qquad H_{k\pm} = H_k[g = g_0 \pm g_1], \tag{11}$$

where $H_{kF}$ denote the Floquet Hamiltonian for the momentum mode $k$ and $H_F = \sum_{k>0} \psi_k^\dagger H_{kF} \psi_k$. A straightforward computation leads to the exact Floquet Hamiltonian $H_{kF}$ given by [58]

$$H_{kF} = \theta_k(T)(\vec{\tau} \cdot \vec{n}_k), \qquad \theta_k(T) = \frac{1}{T}\arccos\left(\cos\phi_k^+ \cos\phi_k^- - (\vec{p}_k^- \cdot \vec{p}_k^+)\sin\phi_k^+ \sin\phi_k^-\right),$$

$$E_k^\pm = \sqrt{(g_0 \pm g_1 - \cos k)^2 + \kappa_0^2 \sin^2 k}, \qquad \phi_k^\pm = E_k^\pm T/2,$$

$$p_k^\pm = \left(0, \sin\Delta_k^\pm, \cos\Delta_k^\pm\right)^T, \qquad \Delta_k^\pm = \arccos\left(\frac{g_0 \pm g_1 - \cos k}{E_k^\pm}\right), \tag{12}$$

where the unit vector $\vec{n}_k$ is given by

$$
\begin{aligned}
n_k^y &= \frac{\kappa_0 \sin k}{\sin(T\theta_k(T))} \sum_{s=\pm} \frac{\sin\phi_k^s \cos\phi_k^{\bar{s}}}{E_k^s}\,, \\
n_k^x &= -\frac{2g_1\kappa_0 \sin k}{\sin(T\theta_k(T))} \frac{\sin\phi_k^+ \sin\phi_k^-}{E_k^+ E_k^-}\,, \\
n_k^z &= \frac{1}{\sin(T\theta_k(T))} \sum_{s=\pm} \frac{(g_0 + sg_1 - \cos k)\sin\phi_k^s \cos\phi_k^{\bar{s}}}{E_k^s}\,,
\end{aligned}
\tag{13}
$$

where $\bar{s} = \mp$ for $s = \pm$. We note that for $T \to 0$, $T\theta_k(T), n_k^x \to 0$ and $H_F$ reduces to its time averaged value as is expected from the first term of a high-frequency Magnus expansion in $T$.

Having obtained an expression for $H_k^F$, the evolution operator $U_k(mT, 0)$ for any momentum mode $k$ after $m$ drive cycles can be obtained as

$$
U_k(mT, 0) = e^{-imT\theta_k(T)(\vec{\tau} \cdot n_k)} = (I\cos(m\theta_k T) - i(\vec{\tau} \cdot \vec{n}_k)\sin(m\theta_k T))\,,
\tag{14}
$$

where $I$ denotes the $2 \times 2$ identity matrix.

Next, we note that for any initial state,

$$
|\psi\rangle_{\text{in}} = \prod_{k>0} |\psi_k^0\rangle\,, \qquad |\psi_k^0\rangle = \left(u_k^0 + v_k^0 c_k^\dagger c_{-k}^\dagger\right)|\text{vac}\rangle\,,
\tag{15}
$$

$$
u_k^0 = \sin(\Delta_k^0/2)\,, \qquad v_k^0 = \cos(\Delta_k^0/2)\,, \qquad \sin\Delta_k^0 = \frac{\kappa_0\sin k}{\sqrt{(g_0 - \cos k)^2 + (\kappa_0\sin k)^2}}\,,
$$

where $|\text{vac}\rangle$ denotes fermionic vacuum and $\kappa_0$ indicates the degree of symmetry breaking parameter. The driven state after $m$ drive cycles can be written as

$$
|\psi_k\rangle = \left(u_k(mT) + v_k(mT)c_k^\dagger c_{-k}^\dagger\right)|\text{vac}\rangle\,, \qquad \begin{pmatrix} u_k(mT) \\ v_k(mT) \end{pmatrix} = U_k(mT, 0)\begin{pmatrix} u_k^0 \\ v_k^0 \end{pmatrix}.
\tag{16}
$$

The coefficients $u_k(mT)$ and $v_k(mT)$ can be computed using Eqs. 14 and 15 and are given, up to an unimportant global phase, by

$$
\begin{aligned}
u_k(mT) &= \sin(\eta_k(mT)/2)\,, \\
v_k(mT) &= e^{i\gamma_k(mT)}\cos(\eta_k(mT)/2)\,, \\
\sin(\eta_k(mT)/2) &= \left[\left(\cos(mT\theta_k(T))u_k^0 - n_k^y\sin(mT\theta_k(T))v_k^0\right)^2\right. \\
&\qquad \left. + \sin^2(mT\theta_k(T))\left(n_k^z u_k^0 + n_k^x v_k^0\right)^2\right]^{1/2}\,, \\
\gamma_k(mT) &= \arccos\left[\cos(mT\theta_k(T))v_k^0 + n_k^y\sin(mT\theta_k(T))u_k^0\right)/\cos(\eta_k(mT)/2)\right] \\
&\qquad - \arccos\left[\cos(mT\theta_k(T))u_k^0 - n_k^y\sin(mT\theta_k(T))v_k^0\right)/\sin(\eta_k(mT)/2)\right],
\end{aligned}
\tag{17}
$$

where we have chosen $u_k^0$ and $v_k^0$ to be real. Using Eqs. 17 and 10, one can compute the fermionic correlation matrix after $m$ drive cycles as

$$
G_k(mT) = \begin{pmatrix} \cos\eta_k(mT) & -ie^{-i\gamma_k(mT)}\sin\eta_k(mT) \\ ie^{i\gamma_k(mT)}\sin\eta_k(mT) & -\cos\eta_k(mT) \end{pmatrix}\,,
\tag{18}
$$

where we have neglected an unimportant constant term.

To obtain $\Delta S_n(mT)$ (Eq. 9), we follow the analysis of Ref. [11]. For an integrable model such as XY chain, the elements of the real space correlation matrix $\Gamma$ for a subsystem $A$ of dimension $\ell$ may be expressed in terms of the correlation matrix $G_k(mT)$ as

$$
\Gamma_{jj'}(mT) = \int_{-\pi}^{\pi} \frac{dk}{2\pi} G_k(mT)e^{ik(j-j')}\,, \qquad j, j' \in A.
\tag{19}
$$

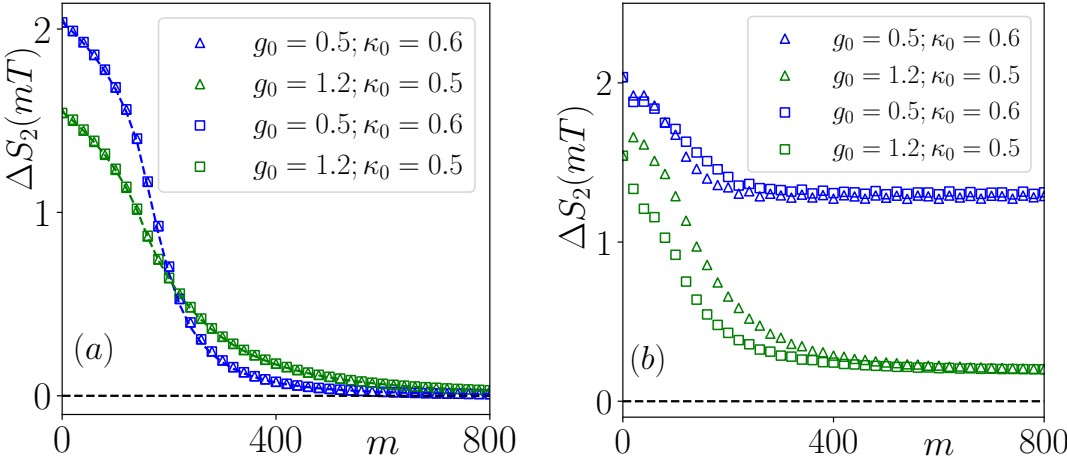

Figure 1: (a) Plot of $\Delta S_2(mT)$ for $\hbar\omega_D/J = g_1 = 20$ as a function of the number of drive cycles $m$ showing symmetry restoration at late times. The green triangles (squares) correspond to results from first-order perturbation theory (exact numerics) for $g_0 = 0.5$ and $\kappa_0 = 0.6$; the corresponding blue symbols show similar plots for $g_0 = 1.2$ and $\kappa_0 = 0.6$. The dotted line corresponds to results from the quasiparticle picture using Eq. 24 The crossing of the two curves indicate presence of quantum Mpemba effect. (b) Same as in (a) but for $\hbar\omega_D/J = 3g_1/4 = 15$; here dynamical symmetry restoration is absent at late stroboscopic times (large $m$) and first order perturbation theory slightly deviates from the exact result for $g_0 = 1.2(0.5)$ and $\kappa_0 = 0.5(0.6)$. For all plots, $L = 4000$, $\ell = 100$, and $J = 1$. See text for details.

In terms of $2\ell \times 2\ell$ matrix $\Gamma(mT)$ one can compute [11]

$$Z_n(\boldsymbol{\alpha}; mT) = \sqrt{\det\left[\left(\frac{I-\Gamma}{2}\right)^n\left(I_\ell + \prod_{p=1}^{n}(I+\Gamma)(I-\Gamma)^{-1}e^{i(\alpha_{p+1}-\alpha_p)\tilde{n}}\right)\right]}, \qquad (20)$$

where $I_\ell$ denotes the $2\ell \times 2\ell$ identity matrix and $\tilde{n}$ is a diagonal $2\ell \times 2\ell$ matrix whose elements are $-1$ for odd row odd column and $1$ for even row even column respectively. It is to be noted that the matrix $(I_\ell - \Gamma)$ is non-invertible and thus evaluation of $Z_n$ requires regularization of this matrix. In our scheme, we carry out this regularization by replacing $(I_\ell - \Gamma)$ by $(\eta I_\ell - \Gamma)$ and taking the limit $\eta \to 1$ at the end of the computation.

Using Eqs. 9, 18, 19 and 20, we numerically compute $\Delta S_n(mT)$ for $n = 2$ as shown in Fig. 1. Comparing Fig. 1(a) and (b), we find that the nature of $\Delta S_2(mT)$ depends qualitatively on the drive frequency. At special frequencies, for which $g_1 J/\hbar\omega_D = p$ with $p \in Z$ (Fig. 1(a)), the driven chain shows dynamical symmetry restoration and $\Delta S_2(mT) \to 0$ for large $m$. In contrast, at the other frequencies (Fig. 1(b)) we do not have such symmetry restoration and $\Delta S_2(mT)$ approaches a finite constant value at large $m$. A plot of

$$\Delta S_2^{\text{avg}} = \sum_{m=m_1}^{m_2} \Delta S_2(mT)/(m_2 - m_1), \qquad (21)$$

as a function of drive frequency for $g_1 = 20$ and $m_2(m_1) = 800(700)$, shown in Fig. 2(a), displays prominent dips at $g_1 = p\hbar\omega_D/J$ confirming the presence of symmetry restoration at these frequencies. We find that such a dynamic symmetry restoration is lost at lower drive frequencies or amplitudes; our numerics implies a crossover between the two regimes as a

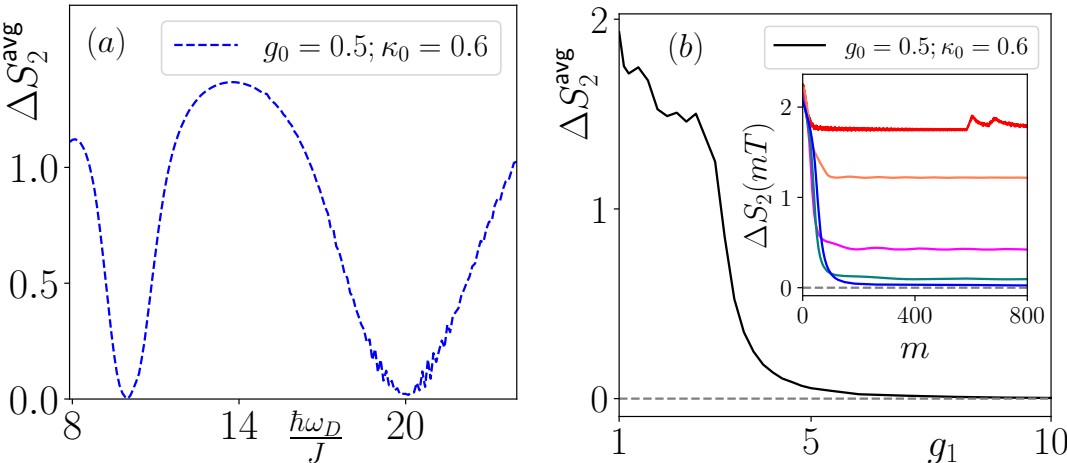

Figure 2: (a) Plot of $\Delta S_2^{\text{avg}}$ for $m_2 = 800$ and $m_1 = 700$ as a function of $\hbar\omega_D/J$ for $g_0 = 0.5$, $\kappa_0 = 0.6$ and $g_1 = 20$ showing dynamical symmetry restoration at special drive frequencies for which $g_1 = p(\hbar\omega_D/J)$ with $p \in Z$. The dips in the figure correspond to $p = 1$ and 2. (b) Plot of $\Delta S_2^{\text{avg}}$ as a function of drive frequency $g_1$ with $g_1 = \hbar\omega_D/J$. The plot shows the absence of dynamical symmetry restoration at smaller $g_1$ indicating a crossover between the two regimes. The inset shows plots of $\Delta S_2(mT)$ as a function of $m$ for $g_1 = \hbar\omega_D/J = 6$ (blue), 4.5 (green), 3.5 (pink), 3.0 (yellow), and 1.5 (red). For all plots, $L = 4000$, $\ell = 100$, and $J = 1$. See text for details.

function of $g_1$ with $\hbar\omega_D/J = g_1$ as shown in Fig. 2(b). The inset of Fig. 2(b) shows the change in behavior of $\Delta S_2(mT)$ as one moves to lower drive amplitudes; we find that $\Delta S_2(mT)$ approaches a finite value at large $m$ which increases with decreasing $g_1$ in this regime.

To understand the reason for such dynamic symmetry restoration, we now turn to the Floquet Hamiltonian in the regime of large drive amplitude $g_1 \gg g_0, 1$. In this regime, it is easy to see from Eqs. 12 and 13 that $p_\pm \simeq (0,0,1)$ and $\phi_k^\pm \simeq \phi_k = g_1 T/2$. This leads to $\theta_k \simeq 2\phi_k$; for $g_1 T = 2p\pi$ where $\sin\phi_k^\pm = 0$, we find from Eq. 13 that $n_k^z \simeq 1$ and $n_k^{x,y} \simeq 0$ for all $k$. This indicates that the leading term in the Floquet Hamiltonian $H_F \sim \tau^z$ and hence $[\tau^z, H_F] = 0$ to leading order. We note that a finite value of $\phi_k^+ - \phi_k^-$, which necessarily contributes to $H_F$ to $O(1/g_1)$ spoils this conservation; thus the emergent symmetry is approximate.

To see this approximate emergent symmetry a bit more clearly, we provide a perturbative derivation of $H_{kF}$ for large drive amplitudes. To this end we obtain the exact evolution operator, $U_0(t)$, corresponding to the largest term $H_{0k} = g_1 \tau^z$ in the Hamiltonian. This yields, for $g_1 \gg g_0, 1$ (we have set $J = 1$)

$$
\begin{aligned}
U_{0k}(t,0) &= e^{itg_1\tau^z/\hbar}, && t \leq T/2, \\
&= e^{i(T-t)g_1\tau^z/\hbar}, && T/2 < t \leq T.
\end{aligned}
\tag{22}
$$

Since $U_{0,k}(T,0) = I$, one finds $H_{kF}^{(0)} = 0$. The effect of the other terms in $H_k(t)$ can be obtained perturbatively. A straightforward calculation, using Floquet perturbation theory yields (for $J = 1$) [52]

$$
U_{1k}(T,0) = -iH_{kF}^{(1)}T/\hbar = \left(-\frac{i}{\hbar}\right)\int_0^T dt\, U_{0k}^\dagger(t,0)(H_k - H_{k0})U_{0k}(t,0),
$$

$$
H_{kF}^{(1)} = \left[(g_0 - \cos k)\tau^z + \kappa_0 \sin k \frac{\sin(g_1 T/(2\hbar))}{g_1 T/(2\hbar)}\left(ie^{-ig_1 T/(4\hbar)}\tau^+ + \text{h.c.}\right)\right],
\tag{23}
$$

where $H_F^{(1)} = \sum_k \psi_k^\dagger H_{kF}^{(1)} \psi_k$. Thus we find that for $g_1 T/\hbar = 2p\pi$, or equivalently $g_1/(\hbar\omega_D) = p$ where $p \in Z$, $[\tau^z, H_{kF}^{(1)}] = 0$ for all $k$. This constitutes an emergent approximate symmetry of the Floquet Hamiltonian which is not respected at higher order [52, 70]. These higher order terms are suppressed by $1/(\hbar\omega_D)$ and therefore for large enough drive amplitude, $H_F^{(1)}$ controls the dynamics. Since stroboscopic time evolution controlled by $H_F^{(1)}$ amounts to time evolution by an effective Hamiltonian which conserves $\tau^z$ (and hence $n_k$), we find dynamical symmetry restoration at these frequencies. This also allows for observation of the quantum Mpemba effect analogous to that in quench dynamics of the model as analyzed in Refs. [11]. However, no such symmetry restoration takes place for $g_1 T/\hbar \neq p\pi$ where the Floquet Hamiltonian does not conserve $n_k$ at any order and $\Delta S_A$ always remain finite. We note that $\Delta S_2(mT)$ obtained using $H_F^{(1)}$ (blue and green triangles in Fig. 1(a)) matches the exact numerics (blue and green squares Fig. 1(a)) qualitatively showing the efficacy of the perturbative expansion at high drive amplitude and special frequencies. Away from the special frequencies, we find a qualitative match between exact numerics and first order perturbative result. The loss of dynamic symmetry restoration at lower drive amplitude (with $g_1/(\hbar\omega_D) = p$) occurs due to the effect of higher order terms in the perturbative expansion which becomes important at low $g_1$.

The behavior of the entanglement asymmetry, in these integrable models, can be further understood in terms of a quasiparticle picture as shown in Ref. [12] for a quench protocol. The idea put forth in Ref. [12] can be generalized in the present case due to the fact that for such integrable models the Floquet Hamiltonian is quadratic in fermionic operators and can be written as sum over its momentum modes. This puts $H_F$ in the same footing with $H$ used for quench for this class of models; thus the analysis carried out in Refs. [11–13] for quench dynamics can be brought to bear in the present case.

The above-mentioned analysis is most easily done at the special drive frequencies where one has dynamic symmetry restoration. To this end, we note that similar to the Hamiltonian $H$ for quench dynamics, $H_F^{(1)}$, which governs the stroboscopic evolution, can act as a source of quasiparticle excitations at any given $k$; the velocity of these quasiparticles are given, in terms of the Floquet quasienergies (Eq. 23) by $v_F(k) = \sin k$ at the special frequencies where $g_1 T/\hbar = 2p\pi$. It turns out that one can compute $Z_n(\alpha; mT)$ by counting the number of such quasiparticles generated over $m$ cycles are shared between subsystems A and B. Using the formalism developed in Refs. [11–13], one finds

$$\Delta S_2(mT) = -\int_{-\pi}^{\pi} \frac{d\alpha}{2\pi} e^{(A_2(\alpha) + B_2(\alpha, mT))\ell},$$

$$A_2(\alpha) = \int_{-\pi}^{\pi} \frac{dk}{4\pi} \ln\left[f(\cos\Delta_k, \alpha) f(\cos\Delta_k, -\alpha)\right], \tag{24}$$

$$B_2(\alpha, mT) = -\int_{-\pi}^{\pi} \frac{dk}{4\pi} \xi(k, \ell, mT) \ln\left[f(\cos\Delta_k, \alpha) f(\cos\Delta_k, -\alpha)\right],$$

where $f(y, b) = \cos b + iy \sin b$, $\xi(k, \ell, mT) = \text{Min}(2mT|v_F(k)|/\ell, 1)$ counts the number of excitations contributing to $\Delta S_2$ in an interval $\ell$ after $m$ cycles [12], and $\cos\Delta_k = (g_0 - \cos k)/\sqrt{(g_0 - \cos k)^2 + \kappa_0^2 \sin^2 k}$ provides information about the initial state.

A plot of $\Delta S_2(mT)$ as a function of $m$ is shown via the dotted lines Fig. 1(a); they match remarkably well with the exact numerics. These confirm that at the special frequency, $H_F^{(1)}$ (Eq. 23) controls the dynamics over a long timescale; moreover, the quasiparticle picture correctly reproduces the characteristics of $\Delta S_2(mT)$ in this regime.

# 3 Driven Rydberg chain

In this section, we shall study the entanglement asymmetry for a chain of Rydberg atoms in the presence of a periodic drive. The effective description of these atoms can be achieved in terms of two states on every site $j$ of the chain; these are the ground state $|g_j\rangle$ and the Rydberg excited states $|r_j\rangle$. The number of Rydberg excitations $\hat{n}_j$ on any site $j$ can then be described by a Pauli spin operator $\sigma_j^z$ as $\hat{n}_j = (1 + \sigma_j^z)/2$. The coupling between these two states is controlled in a standard experiment by a two-photon process; the effect of such a process can be described using the operator $\sigma_j^x = (|g_j\rangle\langle r_j| + \text{h.c.})$. Moreover, these atoms, when excited to their Rydberg state, experience a repulsive van-der Walls interaction which decays as $1/x^6$ with the distance $x$ between them; in the strong interaction regime such an interaction may preclude the presence of two Rydberg excitations within a definite distance $y$. This phenomenon is called Rydberg blockade with $y$ being the blockade radius. [82–91].

The low-energy effective Hamiltonian for these atoms is given by [83, 84]

$$H_{\text{Ryd}} = \sum_j \left( w\sigma_j^x - \Delta'\hat{n}_j + V_0 \sum_{j'} \frac{\hat{n}_j \hat{n}_{j'}}{|j - j'|^6} \right), \qquad (25)$$

where $\Delta'$ acts as local chemical potential for Rydberg excitations and is denoted as detuning. In what follows, we shall study these atoms in the regime where $V_0 \gg \Delta', w \gg V_0/2^6$, so that the effect of the interaction can be implemented by blocking the presence of two neighboring Rydberg excitations. This can be achieved by using a local projection operator $P_j = (1 - \sigma_j^z)/2$ which projects an atom to the ground (spin $\downarrow$) state; a spin-flip from $|\downarrow\rangle$ to $|\uparrow\rangle$ is allowed on a site $j$ only if the neighboring sites $j \pm 1$ have $\downarrow$ spins. In this limit, the effective Hamiltonian of the system is given by the so-called PXP model in a magnetic field [92–94]

$$H = \sum_j \left( w\tilde{\sigma}_j^x - \Delta\sigma_j^z \right), \qquad \tilde{\sigma}_j^x = P_{j-1}\sigma_j^x P_{j+1}, \qquad (26)$$

where we have ignored a constant term and $\Delta = \Delta'/2$. The effect of the van-der Walls interaction between further neighboring atoms can also be ignored in this regime.

In what follows, we shall periodically drive the Rydberg chain using a square pulse protocol with time period $T$: $\Delta(t) = \Delta_0 + (-)\Delta_1$ for $t \leq (>)T/2$, where $\Delta_1 \gg \Delta_0, w$ is the drive amplitude. In this large drive amplitude regime, it is possible to obtain an analytic, albeit perturbative, expression for the Floquet Hamiltonian of the driven chain [69, 70]. This can be achieved by noting that for large $\Delta_1$, one write $H = H_a(t) + H_b$ where

$$H_a(t) = -(\Delta(t) - \Delta_0)\sum_j \sigma_j^z, \qquad H_b = \sum_j \left( w\tilde{\sigma}_j^x - \Delta_0\sigma_j^z \right). \qquad (27)$$

The evolution operator corresponding to $H_a$ can be written as

$$\begin{aligned} U_a(t,0) &= e^{i\Delta_1 t \sum_j \sigma_j^z/\hbar}, & t &\leq T/2, \\ &= e^{i\Delta_1(T-t)\sum_j \sigma_j^z/\hbar}, & t &> T/2. \end{aligned} \qquad (28)$$

Note that $U_a(T,0) = 1$ leading $H_F^{(0)} = 0$. The first order perturbative Floquet Hamiltonian can then be obtained as using Eq. 23 with $U_0 \to U_a$ and $(H - H_0) \to H_b$. This computation is detailed in Refs. [69, 70] and yields, for the first-order Floquet Hamiltonian,

$$H_F^{(1)} = \sum_j \left( w\frac{\sin x_0}{x_0} \left( \tilde{\sigma}_j^x \cos x_0 - \tilde{\sigma}_j^y \sin x_0 \right) - \Delta_0 \sigma_j^z \right), \qquad x_0 = \Delta_1 T/(2\hbar). \qquad (29)$$

We note that as in Sec. 2, for special drive frequencies at which $x_0 = p\pi$ (where $p$ is an integer), $[H_F^{(1)}, \sigma_j^z] = 0$; this leads to an additional emergent symmetry which controls the dynamics up to a prethermal timescale. This timescale can be exponentially large for large $\Delta_1$ thus making the properties of $H_F^{(1)}$ experimentally relevant [68]. We note that for $x_0 = p\pi$, the dynamics of the spins occur due to higher order terms in $H_F$; however, the presence of the emergent symmetry still controls its nature for a long time scale as we shall see from exact numerics.

To compute $\Delta S_2(mT)$ for the driven chain, we follow Ref. [11] and start from an initial state [11–13]

$$|\psi_0\rangle = e^{i\theta \sum_j \tilde{\sigma}_j^y} |\downarrow\downarrow \dots \downarrow\rangle, \qquad (30)$$

where $\theta$ is a rotation angle. Note that for $\theta = 0$, $|\psi_0\rangle$ is an eigenstate of $\sigma_j^z$; thus $\theta$ represent the degree of symmetry breaking in the initial state. The projected initial density matrix $\rho_Q^{(0)}$ is obtained by numerically projecting $\rho^{(0)} = |\psi_0\rangle\langle\psi_0|$ into different sectors of total magnetization $M = \sum_j \sigma_j^z$.

The computation for $\Delta S_2$ in such non-integrable systems is carried out numerically using exact diagonalization (ED). To this end, we first obtain eigenvalues and eigenvectors of the Hamiltonian $H_a(t \leq T/2) = H_-$ and $H_a(t > T/2) = H_+$ as [69, 70]

$$H_\pm |n_\pm\rangle = \epsilon_n^\pm |n_\pm\rangle. \qquad (31)$$

The evolution operator $U(T, 0)$ can then be written as

$$U(T, 0) = e^{-iH_+ T/(2\hbar)} e^{-iH_- T/(2\hbar)} = \sum_{m_-, n_+} c_{m_- n_+} e^{-i(E_m^- + E_n^+)T/(2\hbar)} |n_+\rangle\langle m_-|, \qquad (32)$$

where $c_{m_- n_+} = \langle n_+ | m_- \rangle$. The eigenstates and eigenvectors of $U(T, 0)$ are then obtained using ED; they are denoted by $|\alpha\rangle$ and $\mu_\alpha = \exp[-i\theta_\alpha]$. This procedure also yields the eigenvectors and the eigenvalues of the Floquet Hamiltonian as $|\alpha\rangle$ and $\epsilon_\alpha^F = \arccos[\text{Re}(\mu_\alpha)]$. One can then obtain the state at time $t = mT$ using $|\psi(t)\rangle = U(mT, 0)|\psi_0\rangle$; the reduced density matrix corresponding to subsystem $A$ $\rho_A(mT) = Tr_B[\rho(mT)] = Tr_B[|\psi(mT)\rangle\langle\psi(mT)|]$ can be computed from $|\psi(t)\rangle$ by tracing over rest of the system ($B$) following standard procedure [69, 70]. A similar procedure is carried starting from $\rho_Q^{(0)}$ to obtain $\rho_{QA}(mT)$. Finally, the entanglement asymmetry $\Delta S_2$ is computed from $\rho_A(mT)$ and $\rho_{QA}(mT)$ using Eq. 3 with $n = 2$.

The results obtained from computation of $\Delta S_2(mT)$ for a finite chain of length $L = 24$, subsystem sizes $\ell = 4$, and initial states corresponding to $\theta = \pi/5, \pi/10$ (Eq. 30) is shown in Fig. 3. The left panel shows the evolution of $\Delta S_2$ as a function of the number of drive cycles $m$ away from the special frequencies. We find that in this case, $\Delta S_2 > 0$ for all $m$; moreover it displays rapid oscillations with frequency $\omega_r$ as shown in the inset of Fig. 3(a). The latter feature can be qualitatively understood as Rabi oscillations corresponding to single spin flips occurring during evolution under influence of $H_F^{(1)}$. The frequency of these oscillations matches with the corresponding Rabi frequency $\hbar\omega_r^{(1)} = \sqrt{\Delta_0^2 + (w_0 \sin x_0/x_0)^2}$ (dotted line in the inset of Fig. 3(a)) and is well approximated by $\Delta_0$ for $x_0$ close to $p\pi$. The amplitude of these oscillations varies with $w_0 \sin x_0/x_0$ and decreases as the special frequencies are approached; it vanishes for $x_0 = p\pi$. (Fig. 3(b)).

At the special frequencies, $H_F^{(1)}$ commutes with $\sigma_j^z$ leading to approximate symmetry restoration at long times as long as the higher order terms remain small; consequently, $\Delta S_2(mT) \simeq 0$ for large $m$ at these frequencies (Fig. 3(b)). The dynamics for these special frequencies receives contribution from third order terms which are suppressed by a factor of $1/\omega_D^2$ [69, 70]; this leads to a much longer time scale in dynamics of $\Delta S_2$. Finally, we find from Fig. 3(b), $\Delta S_2(mT)$ shows a faster relaxation for larger $\theta$ leading to realization of the quantum Mpemba effect in a periodically driven constrained quantum system.

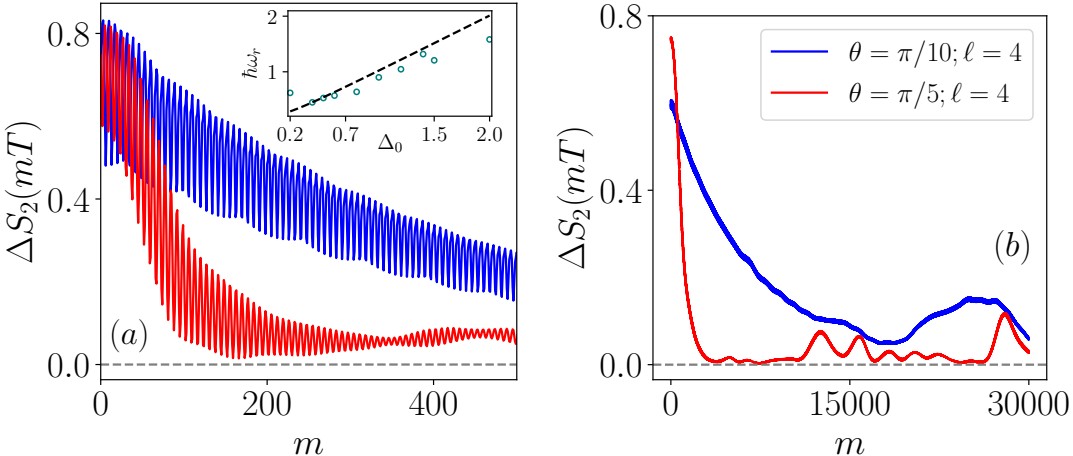

Figure 3: (a) Plot of $\Delta S_2(mT)$ as a function of $m$ for $\ell = 4$ and $\theta = \pi/5, \pi/10$ for $\hbar\omega_D/w = 15$ and $\Delta_1/w = 20$. The plots shows a finite $\Delta S_2$ for all $m$ and rapid oscillations in short time scales. The inset shows the frequency of these oscillations as function of $\Delta$; the dotted line corresponds to the Rabi oscillation frequency $\omega_r^{(1)}$ computed from $H_F^{(1)}$. (b) Similar plots for $\hbar\omega_D = \Delta_1 = 20w$ showing approximate dynamical symmetry restoration over long time scales and absence of fast oscillations. The plots also show faster relaxation for larger $\theta$ thus demonstrating quantum Mpemba effect. For all plots, $L = 24$ and $w = 1$. See text for details.

# 4 CFT on a strip

In this section, we study the dynamics of entanglement asymmetry for a driven CFT on an infinite strip of width $L$ with open boundary condition. To this end, we first define the coordinates of the strip to be $w = \tau + ix$, where the boundary condition is imposed at $x = 0$ and $x = L$, and study the entanglement of a subsystem $A$ of width $\ell$ extending between $x = 0$ to $x = \ell < L$ at $\tau = 0$. In what follows, we define the coordinates of the upper-half plane(UHP) to be $z = \exp[\pi w/L]$ and that of the full complex plane to be $\zeta = \exp[2\pi w/L]$. The results for equilibrium is presented in Sec. 4.1 while those for driven CFTs will be discussed in Sec. 4.2. We shall focus on the computation of the $n^{\text{th}}$ Renyi entropy as the entanglement measure and denote the corresponding entanglement asymmetry as $\Delta S_n$.

## 4.1 Equilibrium result

The computation of $\Delta S_n$ in equilibrium has been carried out for cylinder geometry in several works [25, 27]; more recently, they have been also carried out for a semi-infinite half line [23, 24]. Our motivation for doing this for the strip geometry follows from the fact that in contrast to CFTs on a cylinder, a periodic drive applied to a CFT on a strip allows for nontrivial evolution of the ground state. This is due to the fact that upon mapping the strip into the UHP or an unit disk, such a ground state can always be written as a conformal boundary state. Consequently, it can break the symmetry of the theory and thus change under the time evolution by a Floquet Hamiltonian that we shall consider. This leads to non-trivial dynamics of $\Delta S_n$ which is detailed in Sec. 4.2.

The computation of $S_{Qn}$ can be carried out following a straightforward generalization of obtaining symmetry resolved entanglement using composite twist formalism [21]. To this end,

we first note that one can write $\rho_{QA}^n$ using Eq. 2 as

$$\mathrm{Tr}[\rho_{AQ}^n] = \int_{-\pi}^{\pi} \frac{d\alpha_1' d\alpha_2' \ldots d\alpha_n'}{(2\pi)^n} \mathrm{Tr}\left[\prod_{j=1}^{n} \rho_A e^{-i\alpha_j' Q_A}\right] = \int_{-\pi}^{\pi} \frac{d\alpha_1' d\alpha_2' \ldots d\alpha_n'}{(2\pi)^n} \mathcal{I}(\{\alpha_j'\}), \quad (33)$$

where $\alpha_j' = (\alpha_j - \alpha_{j+1})$, $\alpha_{n+1} = \alpha_1$, and $\sum_{j=1}^{n} \alpha_j' = 0$. Here $Q_A$ denotes the charge operator in subsystem $A$; it is related to the total charge $Q$ by $Q = Q_A + Q_B$ where $B$ denotes the rest of the system. We note here that the initial density matrix $\rho$ for a CFT on a strip corresponds to a boundary conformal state; consequently $[\rho, Q] \neq 0$. This in turn ensures that the reduced density matrix does not commute with $Q_A$: $[\rho_A, Q_A] \neq 0$. This situation is to be contrasted with the one studied in Ref. [21] where $[\rho_A, Q_A] = 0$. For the latter case, since $\sum_j \alpha_j' = 0$, Eq. 33 predicts $\mathrm{Tr}[\rho_{AQ}^n] = \mathrm{Tr}[\rho_A^n]$ and $\Delta S_n$ vanishes.

For a CFT, $\rho_A \exp[-i\alpha_j' Q_A]$ has the natural interpretation of a reduced density matrix in the presence of an Abelian flux; more precisely, the computation of $\mathrm{Tr}[\rho_{QA}^n]$ can be achieved by using the replica method [95] in a $n$-sheeted Riemann geometry $\mathcal{R}_n$ which is glued along the cuts in every sheet. For the strip geometry, after a mapping to the UHP, each of these sheets is extended in the UHP and threaded by Aharanov-Bohm fluxes. In terms of the fields living on $\mathcal{R}_n$ such a flux threading can be implemented using the boundary condition

$$\phi_{j+1}(x, \tau = 0^+) = \phi_j(x, \tau = 0^-) \exp\left[-i\alpha_j'\right], \quad (34)$$

for $x \in A$, where $j = 1 \ldots n$ denotes the sheet index. As shown in Ref. [21], such a flux can be inserted by using a local primary operator $\mathcal{P}_j$ living on the end points of subsystem $A$. Since the computation of entanglement is most easily achieved in terms of a twist operators $\mathcal{T}$ which also lives on the boundary of the subsystem, one can compute the reduced projected density matrix using composite twist operator living on the boundary [21].

To compute the entanglement entropy, we now follow Refs. [21, 95] and carry out a uniformizing transformation

$$\xi(z) = \left(\frac{z - z_+}{z - z_-}\right)^{1/n}, \qquad z_+ = z_-^* = e^{i\pi\ell/L}, \quad (35)$$

which maps $\mathcal{R}_n$ to a disk (which is conformally (global) equivalent to the UHP); the endpoints $z_\pm$ are mapped to $z_+ \to 0$ and $z_- \to \infty$. Note that the strip coordinates $w$ are related to $z$ by the standard relation $z = e^{\pi w/L}$. The stress energy on $\mathcal{R}_n$ can therefore be written as

$$\langle T(z) \rangle_{\mathcal{R}_n, \alpha} = \sum_{j=1 \ldots n} \left(\frac{d\xi}{dz}\right)^2 \langle T_j(\xi) \rangle_{\mathrm{disk}, \alpha} + \frac{cn}{12} \{\xi, z\}_s, \quad (36)$$

where $\{\xi, z\}_s = \xi'''/\xi' - 3/2(\xi''/\xi')^2$ is the Schwarzian derivative and the prime denotes differentiation with respect to $z$. In Eq. 36, the expectation is taken with respect to state in the presence of the flux; since these correspond to states having insertion of local flux via primary operators, these expectation values can be easily computed. The index $\alpha$ indicates the presence of the Aharanov-Bohm flux $\alpha_j'$ between the replicas $j$ and $j+1$ which makes $T_j(\xi)|_{\mathrm{disk}, \alpha}$ non-zero. It is straightforward to compute this contribution using the standard method of images; when $P_j$ has the dimension of $\Delta(\alpha_j') = \bar{\Delta}(\alpha_j')$, one obtains [21]

$$\langle T_j(\xi) \rangle_{\mathrm{disk}, \alpha} = \Delta(\alpha_j')/\xi^2(z), \qquad \{\xi, z\}_s = \frac{n^2 - 1}{2n^2} \frac{(z_+ - z_-)^2}{(z - z_+)^2 (z - z_-)^2},$$

$$\langle T(z) \rangle_{\mathcal{R}_n, \alpha} = d_n \frac{(z_+ - z_-)^2}{(z - z_+)^2 (z - z_-)^2}, \qquad d_n = c(n - 1/n)/24 + \sum_{j=1}^{n} \Delta(\alpha_j')/n^2. \quad (37)$$

Having obtained $\langle T \rangle$, one can now use the standard operator product expansion of the twist operators $\mathcal{T}$ with the stress energy tensor on the UHP. A straightforward calculation, following Refs. [21,95], yields (after identifying $\bar{z} = z^*$)

$$\langle \mathcal{T}(w) \rangle_\alpha = \left( \frac{\partial w}{\partial z} \right)^{-d_n} \left( \frac{\partial \bar{w}}{\partial \bar{z}} \right)^{-d_n} \left( \frac{2i}{z_+ - z_-} \right)^{2d_n} = \left( \frac{L}{\pi a} \sin(\pi \ell / L) \right)^{-2d_n} = \mathcal{I} \left( \{ \alpha_j' \} \right), \quad (38)$$

where $a$ is an ultraviolet cutoff [95]. Note that in the absence of projection to definite symmetry sector, $\Delta(\alpha_j') = \Delta_j(0) = 0$ and Eq. 38 reduces to the well-known result for Renyi entropy on a strip [95]. Substituting Eq. 38 in Eq. 3 we find, after a few lines of algebra,

$$\Delta S_n = \frac{1}{1-n} \ln \prod_{j=1}^{n} \int_{-\pi}^{\pi} \frac{d\alpha_j'}{2\pi} \left[ \frac{L}{\pi a} \sin(\pi \ell / L) \right]^{-2\Delta(\alpha_j')/n^2}. \quad (39)$$

To make further progress, we need to know the dependence of $\Delta$ on the parameter $\alpha$. To this end, we note that such a dependence is given by $\Delta(\alpha) = c_0 \alpha^2 / 2$ for a large class of CFTs [21,26]. These include $c = 1$ Abelian CFTs representing massless boson or free fermions on a chain; for such CFTs the vertex operator corresponding to $U(1)$ symmetry is given by $P_j = \exp[i\alpha\phi_j/(2\pi)]$ and $\Delta = K\alpha^2/(4\pi^2)$. A similar dependence occurs for the $SU(2)_k$ Wess-Zumino-Witten (WZW) models which describes critical spin $k/2$ chains; here the vertex operator is given by $P_j = \exp[i\alpha S_j^z]$, where $S_j^z$ is the $z$ component of the spin, leading to $\Delta = k\alpha^2/(16\pi^2)$. Substituting such a quadratic form of $\Delta$ in Eq. 39 we find that $\Delta S_n$ can be analytically computed (see App. A) for any $n$ and the leading order term, for $a \ll \ell \ll L$, is given by

$$\Delta S_n = \frac{1}{2} \ln \ln \frac{c_0 \ell}{a n^2}, \quad (40)$$

We note that this result exactly matches with the leading term of $\Delta S_n \sim \frac{1}{2} \ln(\ln(\ell/a))$ computed for a interval on the semi-infinite half line in Ref. [24].

## 4.2 Driven CFTs

### 4.2.1 General results

In this subsection, we obtain an expression for $\Delta S_n$ for a CFT driven periodically by an arbitrary drive protocol characterized by a time period $T$. To this end, we consider a generic time evolution operator $U(T_0, 0)$ in Euclidean time with $T_0 = iT$, which, after $m$ cycles of the drive, leads to a coordinate transformation

$$U(mT_0, 0) = e^{-H_F mT_0/\hbar} = \begin{pmatrix} \tilde{a}_m & \tilde{b}_m \\ \tilde{c}_m & \tilde{d}_m \end{pmatrix}, \qquad \zeta \to \zeta_m = \frac{\tilde{a}_m \zeta + \tilde{b}_m}{\tilde{c}_m \zeta + \tilde{d}_m}, \qquad \tilde{a}_m \tilde{d}_m - \tilde{b}_m \tilde{c}_m = 1, \quad (41)$$

on the complex plane. Here we have used the fact that the holomorphic generators $L_0$ and $L_1$ and $L_{-1}$, which constitutes the CFT Hamiltonian, allow a $SU(1,1)$ representation in terms of standard Pauli matrices and the coefficients $\tilde{a}_m, \tilde{b}_m, \tilde{c}_m, \tilde{d}_m$ are functions of $T_0$. A similar transformation can be done for $\bar{\zeta}$.

To implement the effect of the drive, we adapt the procedure of Ref. [71] where one uses a two step conformal transformation. The first maps the strip on the full complex plane with a branch cut where the coordinate transformation corresponding to the drive is implemented. Here one can choose the *conformal boundary condition* just above and below the cut in a way, such that the contribution coming from the operator evolution under the drive can be obtained

by an integral around a closed contour without a branch cut. This can be done as shown in Ref. [71] and leads to, for any primary operator $O(\zeta, \bar{\zeta})$ with conformal dimension $(h, \bar{h})$

$$U(mT_0, 0)^{-1} O\left(\zeta, \bar{\zeta}\right) U(mT_0, 0) = \left(\frac{\partial w}{\partial \zeta}\right)^{-h} \left(\frac{\partial \bar{w}}{\partial \bar{\zeta}}\right)^{-\bar{h}} \left(\frac{\partial \zeta_m}{\partial \zeta}\right)^{h} \left(\frac{\partial \bar{\zeta}_m}{\partial \bar{\zeta}}\right)^{\bar{h}} O\left(\zeta_m, \bar{\zeta}_m\right). \quad (42)$$

The second transformation maps the $O$ to UHP through the transformation $z = \sqrt{\zeta_m}$ and the identification $\bar{z} = z^*$ where all operator expectations values are evaluated. Using this and Cardy's method of images [95], one obtains, after a straightforward calculation detailed in Ref. [71],

$$\langle \mathcal{T}_j(z) \rangle_{\mathrm{UHP}, \alpha} = \left(\frac{\partial w}{\partial \zeta}\right)^{-h_j} \left(\frac{\partial \bar{w}}{\partial \bar{\zeta}}\right)^{-h_j} \left(\frac{\partial \zeta_m}{\partial \zeta}\right)^{h_j} \left(\frac{\partial \bar{\zeta}_m}{\partial \bar{\zeta}_m}\right)^{h_j} \left(\frac{1}{4|\zeta_m|}\right)^{h_j} \left(\frac{2i}{\sqrt{\zeta_m} - \sqrt{\zeta_m^*}}\right)^{2h_j}, \quad (43)$$

$$h_j = c(n - 1/n)/24 + \Delta_j/n^2.$$

We note that for $\tilde{a}_m = \tilde{d}_m = 1$ and $\tilde{b}_m = \tilde{c}_m = 0$, Eq. 43 reproduces the results of the previous section with $\zeta = e^{2\pi i \ell/L}$. Furthermore, as noted in Ref. [71], the value of $(\sqrt{\zeta_m} - \sqrt{\zeta_m^*})^{-2h_j}$ may depend on the branch chosen since it's a multivalued function in the complex plane. In fact, this choice is important for protocols where $\tilde{b}_m$ and $\tilde{c}_m$ are purely real in Euclidean time [71]; however, where $\tilde{b}_m$ and $\tilde{c}_m$ are purely imaginary or in cases where they are complex numbers, both branch choices gives identical result in the large $m$ limit. We shall, in the rest of this work, assume that the latter property holds. This happens to be case for the square-pulse protocol studied in the next subsection.

Substituting $\zeta = \exp[2\pi i \ell/L]$, one can obtain an expression for $\langle \mathcal{T}_j(z) \rangle_{\mathrm{UHP}, \alpha}$ in terms of the coefficients in Euclidean time

$$\langle \mathcal{T}_j(z) \rangle_{\mathrm{UHP}, \alpha} = \beta_m^{-(c(n-1/n)/12 + \Delta_j(\alpha)/n^2)}, \qquad \beta_m = \left(\frac{2L}{\pi}\right)^2 |\tilde{X}_m||\tilde{Y}_m| \left(\mathrm{Im}\left[\sqrt{\tilde{X}_m \tilde{Y}_m^*}\right]\right)^2, \quad (44)$$

$$\tilde{X}_m = \left(\tilde{a}_m e^{2\pi i \ell/L} + \tilde{b}_m\right), \qquad \tilde{Y}_m = \left(\tilde{c}_m e^{2\pi i \ell/L} + \tilde{d}_m\right).$$

Next, we Wick rotate to real time: $T_0 \to iT$, where $T$ is the time period of the drive. This leads to $\tilde{a}_m \to a_m = \tilde{a}_m(T_0 \to iT)$ and similar changes for other coefficients. These coefficients satisfy $a_m = d_m^*$ and $b_m = c_m^*$. We then define $X_m = (a_m e^{2\pi i \ell/L} + b_m)$, and $Y_m = (c_m e^{2\pi i \ell/L} + d_m)$, and consider $\Delta_j(\alpha) = c_0 \alpha^2$ as done in Sec. 4.1. Repeating computations analogous to those in Sec. 4.1 and A, we find the leading contribution to $\Delta S_n(mT)$ for large $m$, is given by

$$\Delta S_n(mT) \simeq \frac{1}{2} \ln \gamma_m, \qquad \gamma_m = \frac{c_0 \ln \beta_m}{n^2}. \quad (45)$$

Eq. 45 constitutes an analytic expression for the entanglement asymmetry for a driven CFT in a strip geometry after $m$ cycles of the drive with period $T$.

### 4.2.2 Square-pulse protocol

To make further progress we choose a concrete drive protocol. In what follows, we consider a square-pulse drive protocol given by

$$\begin{aligned} H &= 2L_0, & \text{for } t \leq T_1, \\ &= 2L_0 + i\mu_0(L_1 - L_{-1}), & \text{for } T_1 < t \leq T_2, \end{aligned} \quad (46)$$

where $T = (T_1 + T_2)$ is the drive period and $\mu_0$ is a real parameter. These holomorphic generators obey $su(1,1)$ subalgebra and hence have the representation in terms of standard Pauli matrices $\sigma_{\alpha=x,y,z}$ as

$$L_0 = \sigma_z/2, \qquad L_1 = -\sigma_-, \qquad L_{-1} = \sigma_+, \quad (47)$$

which allows one to write

$$
\begin{aligned}
H &= \sigma_z, && \text{for } t \le T_1, \\
&= \sigma_z - i\mu_0 \sigma_x, && \text{for } T_1 < t < T_2.
\end{aligned}
\tag{48}
$$

We note that for Euclidean evolution using the protocol given by Eq. 48, in Euclidean time the coefficients $\tilde{a}_m$ and $\tilde{d}_m$ (Eq. 41) are real valued functions of $T_0$ while $\tilde{b}_m$ and $\tilde{c}_m$ are purely imaginary. In contrast, in real time, after one drive cycle $U(T,0)$ is given by

$$
U(T,0) = e^{-iH_F T} = \begin{pmatrix} a & b \\ c & d \end{pmatrix},
\tag{49}
$$

which yields complex valued coefficients with $a = d^*$ and $b = c^*$. For $\mu_0^2 > 1$, $\mu_0^2 < 1$ and $\mu_0^2 = 1$ we have three different sets of $a, b$ as the following:

$$
\begin{aligned}
a &= e^{-iT_1}\left( \cos \nu T_2 - \frac{i}{\nu} \sin \nu T_2 \right), \\
b &= -e^{iT_1}\frac{\mu_0}{\nu} \sin \nu T_2,
\end{aligned}
\qquad \text{where } \nu = \sqrt{1-\mu_0^2}, \quad \mu_0^2 < 1,
\tag{50}
$$

$$
\begin{aligned}
a &= e^{-iT_1}\left( \cosh \eta T_2 - \frac{i}{\eta} \sinh \eta T_2 \right), \\
b &= -e^{iT_1}\frac{\mu_0}{\eta} \sinh \eta T_2,
\end{aligned}
\qquad \text{where } \eta = \sqrt{\mu_0^2 - 1}, \quad \mu_0^2 > 1,
\tag{51}
$$

$$
\begin{aligned}
a &= e^{-iT_1}\left( 1 - iT_2 \right), \\
b &= -T_2 e^{iT_1},
\end{aligned}
\qquad\qquad\qquad\qquad \mu_0^2 = 1.
\tag{52}
$$

Note that depending on $\mu_0$, $T$ and $T_1$, one can have $|\mathrm{Tr}\, U| > 2$, $< 2$, or $= 2$. These correspond to heating, non-heating and critical phases the driven CFT respectively; the transition between these phases can be implemented by tuning $T$, $T_1$ and $\mu_0$ and constitutes an example of dynamical transition in driven CFTs [71]. From the expression of $U(T,0)$, we now compute the Floquet Hamiltonian $H_F$ so that it is possible to obtain $U(mT,0)$ given by

$$
U(mT,0) = e^{-iH_F mT} = \begin{pmatrix} a_m & b_m \\ c_m & d_m \end{pmatrix}.
\tag{53}
$$

A straightforward computation allows one to write, for $\mathrm{Tr}\, U < 2$, in the non-heating phase

$$
\begin{aligned}
a_m &= \cos(\theta(T)mT) + i\frac{a_I}{\sin(\theta(T)T)} \sin(\theta(T)mT) = d_m^*, \\
b_m &= \frac{b}{\sin(\theta(T)T)} \sin(\theta(T)mT) = c_m^*.
\end{aligned}
\tag{54}
$$

Here $a_I \equiv \mathrm{Im}(a)$, $a_R \equiv \mathrm{Re}(a)$ and $\theta(T) = \cos^{-1}(a_R)/T$. Similarly for $(\mathrm{Tr}\, U > 2)$, in the heating phase, one obtains

$$
\begin{aligned}
a_m &= \cosh(\theta(T)mT) + i\frac{a_I}{\sinh(\theta(T)T)} \sinh(\theta(T)mT) = d_m^*, \\
b_m &= \frac{b}{\sinh(\theta(T)T)} \sinh(\theta(T)mT) = c_m^*.
\end{aligned}
\tag{55}
$$

Here $\theta(T) = \cosh^{-1}(a_R)/T$. The expressions for $a$ and $b$ are given in (51). For the phase boundary $(\mathrm{Tr}\, U = 2)$, we find

$$
a_m = 1 + ima_I = d_m^* \quad \text{and} \quad b_m = mb = c_m^*.
\tag{56}
$$

In the next subsection, we shall use Eqs. 54, 55, and 56 to study the time variation of $\Delta S_n$.

### 4.2.3 Behavior of $\Delta S_n(mT)$

We begin by substituting Eqs. 54, 55, and 56 in Eq. 44. We obtain, after some algebra,

$$\beta_m = 2(L/(a\pi)^2)|F_m|^2\left(-\Lambda + \sqrt{\Lambda^2 + \delta^2}\right), \tag{57}$$

where

$$|\tilde{X}_m| \to F_m = \left(a_m^2 - b_m^2 - 2ia_m b_m \sin(2\pi\ell/L)\right]\right)^{1/2}, \qquad |\tilde{Y}_m| \to F_m^*, \tag{58}$$

$$\delta = 2\text{Im}\left(b_m a_m^*\right) + \left(|a_m|^2 + |b_m|^2\right)\sin\frac{2\pi l}{L}, \qquad \Lambda = \cos\frac{2\pi l}{L}. \tag{59}$$

Next, we use Eqs. 58 and 57 to compute entanglement asymmetry from Eq. 45 for different phases in the large $m$ limit. In the heating phase, we find

$$\delta|_{mT\to\infty} = \frac{1}{2}e^{2\theta(T)mT}\left[\frac{|b|^2\sin\frac{2\pi l}{L} - b_R a_I + b_I\sinh(\theta(T)T)}{\sinh^2(\theta(T)T)}\right], \tag{60}$$

$$|F_m|^2|_{mT\to\infty} = \frac{1}{4}e^{2\theta(T)mT}|F'|, \tag{61}$$

where $b_R(a_R)$, $b_I(a_I)$ denotes real and imaginary parts of $b(a)$ (Eqs. 50, 51, and 52) and $|F'| \equiv |F'(T, T_1, \mu_0)|$ (which is independent of $m$) is given by

$$
\begin{aligned}
|F'| &= \sqrt{(\text{Re}F')^2 + (\text{Im}F')^2}, \\
\text{Re}F' &= 1 - \frac{a_I^2 + b_R^2 - b_I^2 - 2\sin\frac{2\pi l}{L}\left[a_I b_R + b_I\sinh(\theta(T)T)\right]}{\sinh^2(\theta(T)T)}, \\
\text{Im}F' &= 2\frac{(a_I\sinh(\theta(T)T) - b_R b_I) - \sin\frac{2\pi l}{L}(b_R\sinh(\theta(T)T) - a_I b_I)}{\sinh^2(\theta(T)T)}.
\end{aligned} \tag{62}
$$

Using Eqs. 57, 60, and 62 we find, for $m \to \infty$,

$$\beta_m \to \left(\frac{L}{2\pi}\right)^2 e^{4\theta(T)mT}|F'|, \left(\frac{|b|^2\sin\frac{2\pi l}{L} - b_R a_I + b_I\sinh(\theta(T)T)}{\sinh^2(\theta(T)T)}\right). \tag{63}$$

Thus for large $m$ and for any fixed $T$, $T_1$ and $\mu_0$ in the heating phase, $\Delta S_n(mT)|_{mT\to\infty} \sim \ln(m)$. This indicates that entanglement asymmetry grows logarithmically in this phase. This behavior is to be contrasted with a steady state value of $\Delta S_n(m)$ at large time for the driven XY and Rydberg chains. This difference can be attributed to the contrasting structure of the evolution operators in the two cases. For driven CFTs in the heating phase, $U(T, 0)$ has a $SU(1, 1)$ representation with negative Casimir allowing, for example, for exponential growth of correlation functions; this situation is therefore qualitatively different from, for example, the XY model, where the associated group is compact ($SU(2)$) with positive Casimir leading to oscillatory behavior of correlators.

In contrast, at the phase boundary a large $m$ limit yields

$$\beta_m|_{m\to\infty} \approx 2(L/\pi)^2\left(F_m^2\delta\right), \tag{64}$$

where, using (56), we have

$$\delta|_{m\to\infty} \approx m^2\left(\left(a_I^2 + |b|^2\right)\sin\frac{2\pi l}{L} - 2b_R a_I\right), \tag{65}$$

$$F_m^2 \approx m^2\left[\left(a_I^2 - b_I^2 + b_R^2 - 2a_I b_R\sin 2\pi\ell/L\right)^2 + 4b_I^2\left(b_R - a_I\sin 2\pi\ell/L\right)^2\right]^{1/2} = m^2|F'|.$$

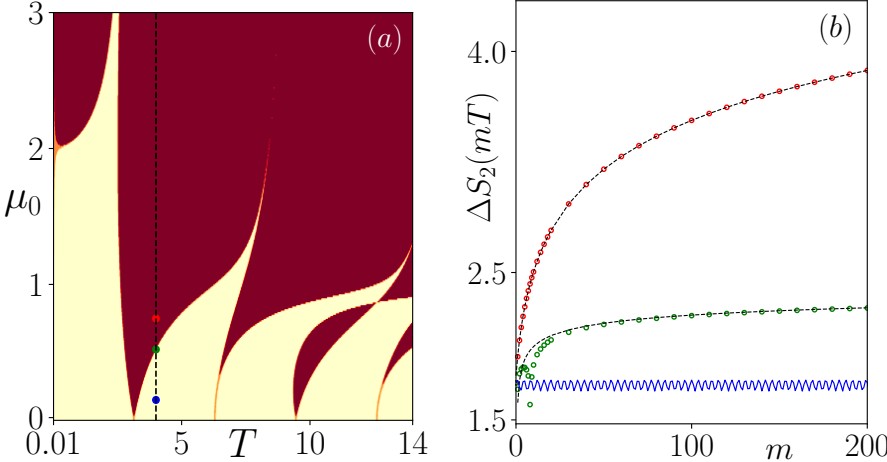

Figure 4: (a) Phase diagram showing the heating (red) and the non-heating (yellow) phases separated by phase boundaries for the driven CFT as a function of $\mu_0$ and $T$ for $T_1 = T/2$. (b) Plot of $\Delta S_2(m)$ as a function of $m$ in the heating phase (red circles) for $\mu_0 = 0.75$ and $T = 4$ (red circle in (a)) showing the $\ln m$ behavior, the phase boundary (green circles) with $T = 4$ and $\mu_0 \simeq 0.5218$ (green circle in (a)) showing $\ln\ln m$ growth, and the non-heating phase (blue line) for $T = 4$ and $\mu_0 = 0.15$ (blue circle in (a)) showing oscillatory behavior. The dashed lines correspond to analytical expressions in the large $m$ limit (Eqs. 63 and 65). For (b) we have chosen $\ell = 100$, $c_0 = 1$, and $L = 1000$.

Here $a$ and $b$ are given by Eq. 52 and their substitution in Eq. 65 yields $\beta_m|_{m\to\infty} \propto m^4$. This allows us to obtain $\Delta S_n(m)|_{m\to\infty} \sim \ln(\ln m)$ leading to a near constant value of $\Delta S$ on the critical line. The non-heating phase, in contrast, do not seem to have a definite large $m$ limit and the behavior of $\Delta S_n(m)$ remains oscillatory at all times.

A numerical representation of our results is summarized in Fig. 4. The phase diagram for the Floquet phases corresponding to the drive given by Eq. 46 is shown in Fig. 4(a) as a function of $\mu_0$ and $T$ for $T_1 = T/2$; the lines corresponding to $|\text{Tr}\,U| = 2$ (Eq. 53) marks the phase boundaries separating the heating ($|\text{Tr}\,U| > 2$, marked in red) and the non-heating ($|\text{Tr}\,U| < 2$, marked in yellow) phases. The corresponding behavior of $\Delta S_2(m)$ (Eq. 45) for different phases is shown in Fig. 4(b). In the heating phase, for $\mu_0 = 0.75$ and $T = 4$, we find the predicted logarithmic growth of $\Delta S_2$ (red circles); the large $m$ analytical result obtained using Eq. 63 for large $m$ is given by the dashed line which fits the numerical curve quite well. The green circles in Fig. 4(b) shows the behavior of $\Delta S_2(m)$ on the phase boundary for $T = 4$ and $\mu_0 \simeq 0.5218$; it indicates a $\ln(\ln m)$ growth for large $m$ as shown using the dotted line (Eq. 65). The blue line shows the behavior of $\Delta S_2$ in the non-heating phase ($T = 4$ and $\mu_0 = 0.15$); we find that in this phase $\Delta S_2$ shows small amplitude oscillations around its initial value which is in sharp contrast to its growth in the other two phases. Thus our results indicate that the behavior of $\Delta S_2$ for a driven CFT depends crucially on the phases of its evolution operator or equivalently Floquet Hamiltonian. Moreover, we find that $\Delta S_2$ exhibits slow growth both in the heating phase and the phase boundary; this feature is qualitatively different from the behavior of its counterpart in typical integrable and non-integrable spin chains studied in earlier sections and also from that in the non-heating phase of the driven CFT.

# 5 Discussion

In this work, we have studied the behavior of entanglement asymmetry, measured using Renyi entropy $\Delta S_n$, for several periodically driven systems. The analysis carried out constitutes a generalization of the earlier studies of entanglement asymmetry to setups with periodic drive. Our results are obtained for both integrable and non-integrable many-body models and for a class of conformal field theories on a strip. Although, we have mainly used second Renyi entropy as measure of entanglement asymmetry for obtaining numerical results in this work, most of these results can be generalized to cases where higher order Renyi or von-Neumann entropies are used as measures.

For integrable models described by free fermions, such as the XY spin chain in a transverse field, our analysis constitutes a generalization of those carried out in Refs. [11–13]; it shows that for periodically driven integrable systems, properties of the entanglement asymmetry depends crucially on the ratio of the amplitude and the frequency of the drive. For the square-pulse protocol studied in this work, at special drive frequencies when this ratio is an integer, the Floquet Hamiltonian of the system hosts an emergent approximate symmetry. In the large drive amplitude regime, the presence of such a symmetry allows for dynamical symmetry restoration leading to $\Delta S_2 \to 0$ at late stroboscopic times. Moreover, at these frequencies, the systems shows quantum Mpemba effect; an initial state with larger degree of broken symmetry relaxes faster to the symmetry restored state. In contrast, away from these special frequencies, such symmetry restoration does not occur and $\Delta S_2$ always remain finite. We have checked that analogous phenomenon occurs for other drive protocols and its signature can be seen when von-Neumann or higher Renyi entropies are used as measures of entanglement asymmetry.

For the non-integrable PXP model, we have numerically shown the existence of analogous symmetry restoration at special drive frequencies using exact diagonalization. Our analysis, carried out for finite systems $L = 24$ and $\ell = 4$ shows the existence of quantum Mpemba effect in these models. We have provided a perturbative semi-analytic explanation of the behavior of the model both at and away from the special frequencies. In the latter regime, as in integrable models, there is no symmetry restoration; interestingly, here the entanglement asymmetry displays fast oscillations whose origin can be qualitatively explained as effect of single spin flips during evolution. The frequency and amplitude of these oscillations match qualitatively with predictions based on $H_F^{(1)}$; these oscillations vanish as the special frequencies are approached.

For conformal field theories on a strip, we have used a composite twist operator formalism where the vertex operators are threaded at the edge of the subsystem. We find a $\ln(\ln \ell/a)$ growth for $a \ll \ell \ll L$ in the strip. This is consistent with results found in Refs. [23,24] for compact Lie symmetry. We have also used our formalism to compute the behavior of entanglement asymmetry for periodically driven CFTS; our analysis shows the long time behavior of $\Delta S_n$ depends on the phase of the driven CFT. We have analytically shown that $\Delta S_n \sim \ln m[\ln(\ln m)]$ in the heating [critical] phase of such driven CFTs on a strip. We note that this indicates qualitatively different behavior of $\Delta S_n$ from both the non-heating phase (where it exhibits small amplitude oscillations around its initial value) and that found for Ising and Rydberg spin chains (where it saturates to a constant value away from the special frequencies and vanishes at the special frequencies).

Our work may lead to several future directions. First, it warrants study of entanglement asymmetry in systems in the presence of quasiperiodic drive protocols [96,97]. It will be useful to understand the behavior of $\Delta S$ in the presence of such drives for both integrable or non-integrable spin chains and CFTs. Second, it will be interesting to look into other non-integrable models such as constrained fermion chains with strong nearest neighbor interactions [98]; it is well known that such systems, under periodic drive, exhibits signatures of Hilbert space fragmentation [99–101]. The behavior of entanglement asymmetry in such systems have not

been studied yet. We leave these topics as possible future studies.

In conclusion, we have studied the behavior of entanglement asymmetry in periodically driven systems and pointed out the role of emergent approximate symmetry of the Floquet Hamiltonian in shaping the behavior of entanglement asymmetry. We have also studied the behavior of $\Delta S$ for driven CFTs on a strip and have shown that it depends crucially on the phase of the driven CFT.

## Acknowledgments

**Funding information** The research work by SD is supported by a DST INSPIRE Faculty fellowship. KS thanks DST, India for support through SERB project JCB/2021/000030.

## A   Computation of $\Delta S_n$ for CFT on a strip

In this section, we briefly outline the computation of $\Delta S_n$ on the strip. We begin with Eq. 39 of the main text and write the $n$-dimensional integral over the variables $\alpha_j$ as

$$I = \int_{-\pi}^{\pi} ... \int_{-\pi}^{\pi} \frac{d\alpha_1...d\alpha_n}{(2\pi)^n} \exp\left[-c_1 \sum_{j=1}^{n} (\alpha_j - \alpha_{j+1})^2 /2\right], \qquad (66)$$

where $c_1 = 2c_0 \ln c(\ell)/n^2$ and $c(\ell) = (L/(\pi a)) \sin(\pi \ell/L)$. To carry out this integral, we write it as

$$I = \int_{-\pi}^{\pi} ... \int_{-\pi}^{\pi} \frac{d\alpha_1...d\alpha_n}{(2\pi)^n} \exp\left[-c_1 \tilde{\alpha}^T M \tilde{\alpha}\right], \qquad (67)$$

where $\tilde{\alpha}^T = (\alpha_1, ....\alpha_n)$ and the non-zero elements of the $n \times n$ dimensional matrix $M$ is given by

$$M_{jj} = 1, \qquad M_{j,j\pm 1} = -1/2, \qquad M_{1,n} = M_{n,1} = -1/2. \qquad (68)$$

It turns out that $M$ is a circulant matrix with real-valued elements and has eigenvalues

$$\epsilon_j = 1 - \cos(2\pi j/n), \qquad j = 1..n. \qquad (69)$$

Its eigenvalues can also be found by noting that $M$ represents a free particle Hamiltonian with nearest neighbor hopping amplitude $t = -1/2$ and chemical potential $\mu = -1$ on a one-dimensional chain.

To compute $I$, we therefore change integration variables to $\tilde{\beta} = U\tilde{\alpha}$, where $U$ is an orthogonal matrix such that $(UMU^T)_{jj'} = \epsilon_j \delta_{jj'}$. Shifting the integration variables to $\beta_j$ and noting that $\epsilon_n = 0$, we find

$$I = \prod_{j=1}^{n-1} \int_{-\pi\mu_j}^{\pi\mu_j} \frac{d\beta_j}{2\pi} \exp\left[-c_1 \epsilon_j \beta_j^2\right], \qquad (70)$$

where $\mu_j$ can be determined using eigenvectors of $M$. Here, we shall be concerned only with the leading order contribution to $I$ when $a \ll \ell \ll L$ where $c_1$ is large. In this limit the integral

$I$ can be written, after rescaling $\beta_j \to \beta_j' = \sqrt{c_1}\beta_j$ as

$$
\begin{aligned}
I &= c_1^{(1-n)/2} \prod_{j=1}^{n-1} \int_{-\pi\sqrt{c_1}\mu_j}^{\pi\sqrt{c_1}\mu_j} \frac{d\beta_j}{2\pi} \exp\left[-\epsilon_j \beta_j^{'2}\right] \\
&= c_1^{(1-n)/2} \prod_{j=1}^{n-1} \frac{\mathrm{Erf}(\pi\sqrt{c_1 \epsilon_j}\mu_j)}{2\sqrt{\pi \epsilon_j}} \\
&\simeq (4\pi c_1)^{(1-n)/2} \prod_{j=1}^{n-1} \epsilon_j^{-1/2},
\end{aligned}
\tag{71}
$$

where in the third line we have taken the limit $c_1 \to \infty$. Note that in this limit the precise expressions of $\mu_j$ do not contribute to the value of $I$. From Eq. 39, we find $\Delta S_n = \ln I/(1-n)$; thus the leading term of $\Delta S_n$ comes from $\ln c_1$ and yields Eq. 40.

Another, equivalent method of evaluating $I$ constitute a shift of variables to $\alpha_j'$. Since $\sum_j \alpha_j' = 0$, one can write

$$
I = \int_{-\pi}^{\pi} \cdots \int_{-\pi}^{\pi} \frac{d\alpha_1' \dots d\alpha_n'}{(2\pi)^n} e^{-c_1 \alpha_j'^2} \delta_{\sum_{j=1}^n \alpha_j', 0}.
\tag{72}
$$

The delta function constraint implements the condition that $\sum_j \alpha_j' = 0$. We implement this constraint using an additional integral over a variable $\lambda$ and write

$$
\begin{aligned}
I &= \int_{-\pi}^{\pi} \frac{d\alpha_n'}{2\pi} e^{-c_1 \alpha_n'^2} \int_{-\pi}^{\pi} \frac{d\lambda}{2\pi} e^{i\lambda \alpha_n'} \left( \int_{-\pi}^{\pi} \frac{d\alpha'}{2\pi} e^{i\lambda \alpha'} e^{-c_1 \alpha'^2} \right)^{n-1} \\
&= \int_{-\pi}^{\pi} \frac{d\alpha_n'}{2\pi} e^{-c_1 \alpha_n'^2} \int_{-\pi}^{\pi} \frac{d\lambda}{2\pi} e^{i\lambda \alpha_n'} e^{-\frac{(n-1)\lambda^2}{4c_1}} I_0^{n-1}, \\
I_0 &= \left( \frac{i}{4\sqrt{\pi c_1}} \right) \left[ \mathrm{Erfi}\left( \frac{\lambda}{2\sqrt{c_1}} - i\pi\sqrt{c_1} \right) - \mathrm{Erfi}\left( \frac{\lambda}{2\sqrt{c_1}} + i\pi\sqrt{c_1} \right) \right].
\end{aligned}
\tag{73}
$$

In the limit $c_1 \to \infty$, we find

$$
\lim_{c_1 \to \infty} I_0 \approx \left( \frac{i}{4\sqrt{\pi c_1}} \cdot (-2i) \right) = (4\pi c_1)^{-1/2},
\tag{74}
$$

and a few lines of algebra yields

$$
\begin{aligned}
I|_{c_1 \to \infty} &\approx (4\pi c_1)^{(1-n)/2} \int_{-\pi}^{\pi} \frac{d\alpha_n'}{2\pi} e^{-c_1 \alpha_n'^2} \int_{-\pi}^{\pi} \frac{d\lambda}{2\pi} e^{i\lambda \alpha_n'} e^{-\frac{(n-1)\lambda^2}{4c_1}} \\
&\approx (4\pi c_1)^{(1-n)/2} \int_{-\pi}^{\pi} \frac{d\alpha_n'}{2\pi} e^{-c_1 \alpha_n'^2} \delta_{(\alpha_n', 0)} \\
&= (4\pi c_1)^{(1-n)/2}.
\end{aligned}
\tag{75}
$$

This yields the same leading order term as obtained from the previous method and leads to Eq. 40 of the main text.

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
