# Peer review of "Entanglement asymmetry in periodically driven quantum systems"

_SciPost Physics, doi:SciPost Phys. 19, 051 (2025)_

## Round 1 · Referee Report · Anonymous (Referee 1) · 2025-2-12

Report

The entanglement asymmetry, which is a useful tool to study the symmetry breaking in the subsystem level, has received extensive attention in the community. This work gives an initial study of entanglement asymmetry in time-dependent driven systems, based on both numerical and analytical methods, in both lattice systems and conformal field theories. The results are novel and interesting. I would recommend the publication of this work after the authors address the following questions/remarks:

— Is there any scaling behavior in the time evolution in Fig.2? Since this is essentially a free-fermion calculation, I think the scaling behavior can be obtained if there is.

— The results in driven spin chains and in driven CFTs are qualitatively different. Is there any physics to understand this difference? Apparently, the driving protocols are very different in these two cases. But it is not clear to me what essential factors result in this qualitative difference in the entanglement asymmetry evolution.

— The analytical results of entanglement asymmetry in driven CFTs are very interesting. I have a technical question here. In Fig.4, the parameters are chosen as l=100 and L=1000. It is known that in driven CFTs, whether one includes the energy density peak(s) in the subsystem or not will give different features in the von Neumann entropy evolution. For the entanglement asymmetry studied here, I wonder if this choice (with energy peak(s) included in the subsystem or not) is still important.

— Since the driven CFT can also be analytically studied when the initial state is a thermal state, as has been recently studied in literature, I think it may be interesting to check how the finite temperature affects the entanglement asymmetry in a driven CFT. This may be an interesting future work to study.

Recommendation

Ask for minor revision

  • validity: high
  • significance: high
  • originality: high
  • clarity: high
  • formatting: excellent
  • grammar: excellent

Author:  Krishnendu Sengupta  on 2025-04-21  [id 5388]

(in reply to Report 1 on 2025-02-12)

See attached file.

Attachment:

main_ref1.pdf

---

## Round 1 · Referee Report · Anonymous (Referee 2) · 2025-3-3

Report

This paper studies the time evolution of the entanglement asymmetry in periodically driven systems. The entanglement asymmetry is a novel quantity that measures the extent to which a symmetry is broken in a portion of a many-body quantum system. As the authors correctly point out, its time evolution has been recently analysed after a global quantum quench in a wide variety of integrable and non-integrable systems, but
there are not works that consider time-dependent periodically driven systems. The present paper fills this gap and explores, combining analytic and numeric methods, the behaviour of the entanglement asymmetry in the periodic driven XY spin chain, in a Rydberg atom chain effectively described by a PXP Hamiltonian with a time dependent periodic magnetic field and in driven CFTs. As in the case of quantum quenches, the authors take an initial state that breaks certain symmetry that is respected by the dynamics. Then they examine whether the symmetry is restored at late times and the occurrence of the quantum Mpemba effect, i.e. the more the symmetry is initially broken, the faster is restored. The authors find a rich phenomenology. For both the XY spin chain and the PXP model, they obtain that, for certain special Floquet frequencies, the symmetry is restored at long times and they observe the quantum Mpemba effect. On the other hand, the entanglement asymmetry exhibits a drastically different behaviour in driven CFTs. Here, instead of decreasing in time, it grows with the number of Floquet cycles, if I correctly understand.

I think that this work presents interesting, novel, and timely results that make it suitable for being published in SciPost. In my opinion, the paper is, in general, well written and complete, except the CFT part. I have several doubts in that section that should be clarified before I can accept it:

i) In Sec. 4.1, the authors seem to mix several setups. They apply the methods from Ref. [21], which studies the resolution of the entanglement entropy in a CFT on the complex plane with respect to a symmetry of the theory. Formula in Eq. (38) seems to assume a cylinder geometry. However, the final result (Eq. (40)) is compared with results from Ref. [24], where the entanglement asymmetry of an interval attached to a boundary that breaks the symmetry is studied. I am confused by the exact setup the authors are considering. This needs to be clarified.

ii) The formulas in Eq. (37) are derived using the Appendix of Ref. [21], where the vertex opertors that implement the conserved charge $e^{-i\alpha Q}$ create a topological defect line. In that case, one can split the flux $\alpha$ among the replicas such that $\sum_{j=1}^n \alpha_j=\alpha$. This mimics the case of asymmetry in Eq. (33). However, the result I obtain for $d_n$ is different from the expression in Eq. (37). I am obtaining $d_n=c(n-1/n)/24+\Delta(\sum_{j=1}^n \alpha_j)/n^2$. What is the crucial point I am missing to get Eq. (37) instead of the $d_n$ I am obtaining?

iii) I think there is a typo in Eq. (38). The last equality should be $=\mathrm{Tr}(\prod_{j=1}^{n-1} \rho_A e^{i\alpha_j Q})$ instead of $=\mathrm{Tr}(\rho_{QA}^n)$, which would be consistent with Eq. (39).

iv) I think Eq. (39) is only valid for Renyi index $n=2$, but it is written $\Delta S_n$.

v) In the first sentence of Sec. 4.1, the authors write "The computation $\Delta S_n$ in equilibrium has been carried out for cylinder geometry in several works [21, 22]". I would like to point out that, in Ref. [21], the entanglement asymmetry is not computed but rather the symmetry-resolved entanglement, which is the opposite situation. I would suggest to cite instead the paper JHEP 05(2024) 059 where the entanglement asymmetry is studied in the ground state of CFTs breaking a symmetry in the bulk and, in particular, in the Ising CFT. Ref. [25] also investigates asymmetry at equilibrium in CFTs in the complex plane.

Apart from the concerns above, I would like to ask the following question:

vi) In the driven XY spin chain and PXP model, does the quantum Mpemba effect always occur when the symmetry is restored? It is not entirely clear to me from the discussion.

Recommendation

Ask for minor revision

  • validity: -
  • significance: -
  • originality: -
  • clarity: -
  • formatting: -
  • grammar: -

Author:  Krishnendu Sengupta  on 2025-04-07  [id 5344]

(in reply to Report 2 on 2025-03-03)

Please see attached file.

Attachment:

main_ref2.pdf

Author:  Krishnendu Sengupta  on 2025-04-07  [id 5343]

(in reply to Report 2 on 2025-03-03)

Please see attached pdf file.

Attachment:

main_ref1.pdf

---

## Round 2 · Referee Report · Anonymous (Referee 1) · 2025-4-21

Report

The authors have resolved the questions in my first report and made the editing accordingly. I recommend the publication of this work of the current form.

Recommendation

Publish (easily meets expectations and criteria for this Journal; among top 50%)

  • validity: -
  • significance: -
  • originality: -
  • clarity: -
  • formatting: -
  • grammar: -

Author:  Krishnendu Sengupta  on 2025-06-12  [id 5563]

(in reply to Report 1 on 2025-04-21)

We thank the referee for their remark and for supporting publication.

---

## Round 2 · Referee Report · Anonymous (Referee 2) · 2025-5-18

Report

I believe the authors have satisfactorily addressed most of my concerns. However, I remain unconvinced that the approach employed to obtain the entanglement asymmetry at equilibrium in a CFT is entirely correct.

In particular, regarding their reply to my question 2, the authors claim that their formula for the dimension of the composite twist operator,
\begin{equation}
\sum_{j=1}^n\frac{\Delta(\alpha_j)}{n^2}
\end{equation}
reduces to the expression by Goldstein and Sela,
\begin{equation}
\frac{\Delta(\alpha)}{n^2}
\end{equation}
when all $\alpha_j$ are equal, i.e. when $\alpha_j=\alpha/n$. However, assuming $\Delta(\alpha)=\alpha^2$ (as the authors do in the paper), we find from their expression
\begin{equation}
\alpha^2/n^3,
\end{equation}
while Goldstein and Sela's formula gives
\begin{equation}
\alpha^2/n^2.
\end{equation}
My proposed formula,
\begin{equation}
\Delta\left(\sum_{j=1}^n\alpha_j\right)/n^2,
\end{equation}
also gives $\alpha^2/n^2$. It predicts when $\sum_j\alpha_j=0$ that $\Delta(\sum_j\alpha_j)=0$, and the asymmetry is zero. This is consistent with the defect being topological. The core issue in Section 4.1 seems to be that the authors assume that the defects are topological in the bulk, but their calculation does not properly take into account that the symmetry is broken in the boundary. As a result, once the correct dimension of the composite twist fields is used, the asymmetry vanishes. To obtain a non-zero asymmetry in this framework, the breaking of the symmetry at the boundary must be explicitly incorporated.

I would also like to remark on a statement after Eq. (34), where the authors write: “In contrast, we demand $[\rho_A,Q_A]\neq 0$; this results in unequal weight of fluxes at every sheet as described in Eqs. 33 and 34.” This statement is, in my view, not correct: In the definition of the asymmetry (Eq. 33), different fluxes $\alpha_j$ are introduced in each replica, regardless of whether $Q_A$ commutes with $\rho_A$. If $\rho_A$ and $e^{i\alpha_jQ_A}$ commute, then the different fluxes cancel and the asymmetry is zero.

While I do not wish to insist further on this calculation, since it is not central to the main results of the manuscript, I would suggest that the authors reconsider this section carefully before the paper is accepted.

Recommendation

Ask for minor revision

  • validity: -
  • significance: -
  • originality: -
  • clarity: -
  • formatting: -
  • grammar: -

Author:  Krishnendu Sengupta  on 2025-06-13  [id 5569]

(in reply to Report 2 on 2025-05-18)

We thank the referee for a careful reading of our manuscript and pointing out some important issues. We have now rewritten parts of the draft to make our calculation method clear and added an appendix. We think that these additions will address referee's concerns. Below, We respond to their comments in details. \

  1. The trace of $n^{\rm th}$ power of the symmetry projected reduced density matrix has the following expression: \begin{align} {\rm Tr}(\rho_{A,Q}^{n}) = \int^\pi_{-\pi} ..... \int^\pi_{-\pi} \frac{d\alpha_1 .... d\alpha_n}{(2\pi)^n} {\rm Tr} \Big[\prod_{j=1}^n \rho_A \exp[i\alpha'_j Q_A] \Big] \end{align} where $\alpha'_j$ =$ \alpha_j-\alpha_{j+1} $ and $\alpha_{n+1}=\alpha_{1}$ so that $\sum_{j=1}^{n}\alpha'_{j} = 0$. In our previous response, we have mistakenly stated that this would reduce to the Goldstein and Sela's answer. We apologize for the misleading statement. We do agree to the referee that our set-up is different to that used by Goldstein and Sela for computing symmetry resolved entanglement entropy. Below, we provide the reason for this. We have now corrected several statements in the manuscript in Secs. 1, 4.1 and 5 and added an appendix to show the details of our calculation.

  2. In our work, we have implicitly assumed that the defect line is not topological. The main justification of this comes from the fact $[\rho,Q]\neq 0$ in our case ( and hence $[\rho_A, Q_A] \ne 0$) since we are working on a strip which hosts boundary state: \begin{align} (T(z)-\bar{T}(\bar{z}))|B\rangle=0 \; \text{at} \; z=\bar{z}. \end{align} A solution of the above equation is a conformal boundary state $|B\rangle$, which can be written in terms of linear combination of Ishibashi states. By definition, an Ishibashi state is constructed out of a primary and taking sum of all level Virasoro descendants acting on that primary with a specific holomorphic structure(for reference, see equation (2.8) of arXiv:1412.6226). This kind of state is expected to break the symmetry of the charge sector since $H|B\rangle \neq 0$. We have computed the Renyi asymmetry in this state. Note that computing $n$-point bulk correlation function in a boundary state is equivalent to $2n$-point holomorphic correlation function in the full complex plane without defect. This `method of image trick' is a consequence of the Ward identity in BCFT, where one need to impose $T(z)=\bar{T}(\bar{z})$ at the boundary(a review of this is given in appendix (D) of arXiv:1907.08763). This further implies in the presence of boundary defect, each Riemann sheet (dressed with $\exp[i \alpha'_j Q_A]$) will contribute to the stress tensor as shown in Eq. (36) of our work and we got (2) as the conformal dimension of composite twist operator.

  3. Regarding how to ensure that the asymmetry vanishes for $[\rho_{A},Q_{A}]=0$, we note the following. As the Referee correctly pointed out, for a topological defect operator, it is possible to fuse the defect lines into the same Riemann sheet (For reference, see JHEP05 (2024) 059). In such a case, the composite twist dimension should reduce to the same as referee has suggested (this can also be seen from expression of $X_n$) and yield \begin{align} d_{n}=\frac{c}{24}(n-\frac{1}{n}) + \Delta(\sum_{j=1}^{n}\alpha'_{j})/n^2 ---- (2) \end{align} Since $\sum_{j=1}^n \alpha'_j=0$, $\Delta=0$ in this case; hence the asymmetry will vanish. This will happen if the initial (ground) state respects the symmetry: $[\rho,Q]=0$. In this case, by definition, $[\rho_{A},Q_{A}]=0$ and we can fuse the defect line into one sheet. Using (3), we can easily see the entanglement asymmetry will vanish in this case. We thank the referee for pointing this out.

  4. We note that from Eqs. (36) and (37) of our work, we end up with \begin{align} d_{n}=\frac{c}{24}(n-\frac{1}{n}) + \sum_{j=1}^{n-1}\Delta(\alpha'_{j})/n^2 \end{align} where $\Delta$ is a function of $\alpha'_j$ i.e. $\Delta(\alpha'_{j})$= $K (\alpha'_{j})^{2}$= $K (\alpha_{j}-\alpha_{j+1})^{2}$ for a class of CFTS. This form is dictated by the fact that for us $[\rho_A, Q_A]\ne 0$. This is central to obtaining the final answer as we claimed in equation (39) and (40) of our work.

We hope that these changes will address referee's concern regarding the computations in Sec. 4.1.

---

## Round 2 · Author Response

Dear Editor

We resubmit as response to the comments by both the referees on our original submission.
All the changes from the earlier version in the main body of the manuscript is colored blue.

Your sincerely

Krishnendu Sengupta

on behalf of
Tista Banerjee, Suchetan Das, and Krishnendu Sengupta

---

## Round 2 · List of Changes

1) We have added a discussion in Sec 4 as response to second comment of Ref 1.

2) We have added a paragraph in the Introduction in response to first comment of Ref 1.

3) We have corrected a typo in Eq 38 of the draft in response to comment 4 of Ref 2

4) We have added Ref 27 in response to comment 5 of Ref 2.

5) We have added a clarification regarding computation of the CFT dynamics in Sec 4.

---

## Round 3 · Referee Report · Anonymous (Referee 2) · 2025-6-24

Report

I thank the authors for the clarifications added in the calculation of the entanglement asymmetry in a strip geometry. I think the manuscript is now ready for publication.

Recommendation

Publish (meets expectations and criteria for this Journal)

---

## Round 3 · Author Response

Dear Editor

We resubmit with our response to the referee. We have made some changes in the manuscript which are color coded; we believe that these will alleviate referee's concern expressed in their previous report.

We hope that the present manuscript will be publishable in Scipost.

Thank you for your time and effort.

Your Sincerely

Tista Banerjee, Suchetan Das and Krishnendu Sengupta

---

## Round 3 · List of Changes

1) We have made changes in Secs 4.1, 1, and 5 in response to regeree's comment. These changes are color coded.

2) We have added App A showing detils of entanglement asymmetry calculation.

---

## Editorial Decision

published